# Collective exchange processes reveal an active site proton cage in bacteriorhodopsin

Daniel Friedrich [1,2,5,6], Florian N. Brünig [3], Andrew J. Nieuwkoop [1,7], Roland R. Netz [3], Peter Hegemann [4] & Hartmut Oschkinat[1,2]*

Proton translocation across membranes is vital to all kingdoms of life. Mechanistically, it relies on characteristic proton flows and modifications of hydrogen bonding patterns, termed protonation dynamics, which can be directly observed by fast magic angle spinning (MAS) NMR. Here, we demonstrate that reversible proton displacement in the active site of bacteriorhodopsin already takes place in its equilibrated dark-state, providing new information on the underlying hydrogen exchange processes. In particular, MAS NMR reveals proton exchange at D85 and the retinal Schiff base, suggesting a tautomeric equilibrium and thus partial ionization of D85. We provide evidence for a proton cage and detect a preformed proton path between D85 and the proton shuttle R82. The protons at D96 and D85 exchange with water, in line with ab initio molecular dynamics simulations. We propose that retinal isomerization makes the observed proton exchange processes irreversible and delivers a proton towards the extracellular release site.

[1] Leibniz-Forschungsinstitut für Molekulare Pharmakologie, Robert-Rössle-Str. 10, 13125 Berlin, Germany. [2] Freie Universität Berlin, Institut für Chemie und Biochemie, 14195 Berlin, Germany. [3] Freie Universität Berlin, Fachbereich Physik, 14195 Berlin, Germany. [4] Humboldt-Universität zu Berlin, Institut für Biologie, Invalidenstr. 42, 10115 Berlin, Germany. [5] Present address: Department of Molecular and Cellular Biology, Harvard University, 52 Oxford Street, Cambridge, MA 02138, USA. [6] Present address: Department of Cancer Biology, Dana-Farber Cancer Institute, 360 Longwood Avenue, Boston, MA 02215, USA. [7] Present address: Department of Chemistry and Chemical Biology, Rutgers, The State University of New Jersey, 123 Bevier Road, Piscataway, NJ 08854, USA. *email: oschkinat@fmp-berlin.de

Bacteriorhodopsin (BR) is an excellent model system to establish new approaches for studying protonation dynamics[1–4]; signals of exchanging protons may be observed in the protein interior for entropic reasons due to their tight coordination within hydrogen bonding networks and limited exchange with bulk water (Fig. 1a, b)[5–7]. The BR pore features three sites with distinct protonation dynamics characteristics, intensely discussed in the literature[4]: the uptake site (D96 and $H_2O$ 502), the active site (D85, D212, R82, retinal Schiff base (RSB), and $H_2O$ 401, 402, and 406), and the proton release group (E194, E204 and a trimeric water cluster ($H_2O$ 403, 404, and 405)) (Fig. 1a). According to the current understanding of the photocycle[8], BR relocates in non-consecutive transport steps a proton across the membrane out of the *Halobacterium salinarum* cell (Fig. 1c)[9–12]. Light reception by the $BR_{568}$ dark-state results in the excited electronic state S1, with the retinal chromophore isomerizing from an all-*trans*,15-*anti* to a 13-*cis*,15-*anti* configuration during the transition to $K_{590}$. In this $K_{590}$ intermediate, however, the chromophore is not fully isomerized yet[13]. The reaction completion requires a subsequent relaxation of retinal into the 13-*cis*,15-*syn* configuration ($L_{550}$ state). This is caused by a drastic $pK_a$ drop of the RSB and an increase of the D85 $pK_a$ accompanied by a proton transition from the RSB to D85, leading to an absorption blue shift to 412 nm ($M1_{412}$)[14]. Protonation of D85 initiates an outward movement of R82 (towards the extracellular site of the protein), a decrease in $pK_a$ of the proton release group and a proton release into the extracellular bulk phase ($L_{550} \rightarrow M1_{412}$)[14–17]. After reorientation of Helix F and large structural changes ($M1_{412} \rightarrow M2_{412}$), water is invading into the proton channel and the RSB is reprotonated from D96 via a transient water chain in the $M2_{412} \rightarrow N_{520}$ transition[18]. Subsequently, protonation of the RSB reduces the $pK_a$ of D85 causing a proton transfer from D85 to the proton release group ($N_{520} \rightarrow O_{640}$). To bridge the long distance between these two sites, this proton relocation step requires additional sites for proton transport, e.g., $H_2O$ molecules or the R82 guanidinium group (Fig. 1a, b). However, it remained unclear whether this late proton transfer involves deprotonation and reprotonation of R82[19]. The photocycle is then completed by restoring the initial all-*trans* configuration of the retinal by thermal relaxation ($O_{640} \rightarrow BR_{568}$)[20]. Key steps in this proton transport process involve $H_2O$ molecules[4,21]. This is especially true for the proton movement from D96 to the RSB but also for the proton release from D85 towards the proton release group (Fig. 1a, b).

Recently, time-resolved studies using serial femtosecond crystallography revealed structural dynamics of retinal isomerization and repositioning of water molecules during the BR photocycle in the active site[2,3]. These studies showed moderate structural changes at the beginning of the photocycle before retinal isomerization, not considering proton relocation independent of heavy atom movements. However, knowledge of preformed proton communication pathways between key sites (e.g., RSB, D85, R82, and the proton release group, the former three forming the active site proton cage) is essential for understanding BR function. Such protonation dynamics within the hydrogen bonding network may involve, for example, proton oscillations at the RSB and D85 as well as hydroxide and hydronium ions in place of proximate water molecules already in the dark-state, which may be indicated by proton nuclear magnetic resonance (NMR) signals with their characteristic chemical shifts. We analyzed the underlying chemical exchange processes site-specifically by magic angle spinning (MAS) NMR and in particular exchange spectroscopy[22]. In this context, the role of R82 in the proton transport mechanism and the nature of water 402 are elucidated in detail because time-resolved infrared spectroscopy for these sites is not available. To this end, we studied deuterated and $^{13}C,^{15}N$-labeled native BR-enriched membranes (purple membranes). Direct proton detection was employed in a temperature range from 100 K to 290 K to monitor protons in the BR dark-adapted state (mixture of 13-*cis*,15-*syn* and 13-*trans*,15-*anti* retinal configurations) at key sites of the proton transport pathway, after introducing a subset of exchanging protons into the interior of the perdeuterated samples upon illumination. It is well-established that the structural rearrangements during the photocycle are required for complete proton transport through the BR pore from the uptake to the release site, which is exploited here for reprotonation of sites involved in the pathway[9,23–25]. Proton-detected MAS NMR exchange spectroscopy enabled the observation of proton localization and displacement, and the underlying exchange. It informs on exchange contributions of the chemical sites involved in the global proton relocation process of BR, even when investigating the dark-adapted equilibrium that potentially features different proton exchange rates compared to the transfer steps during the photocycle. Ab initio molecular dynamics simulations allowed for an estimation of proton

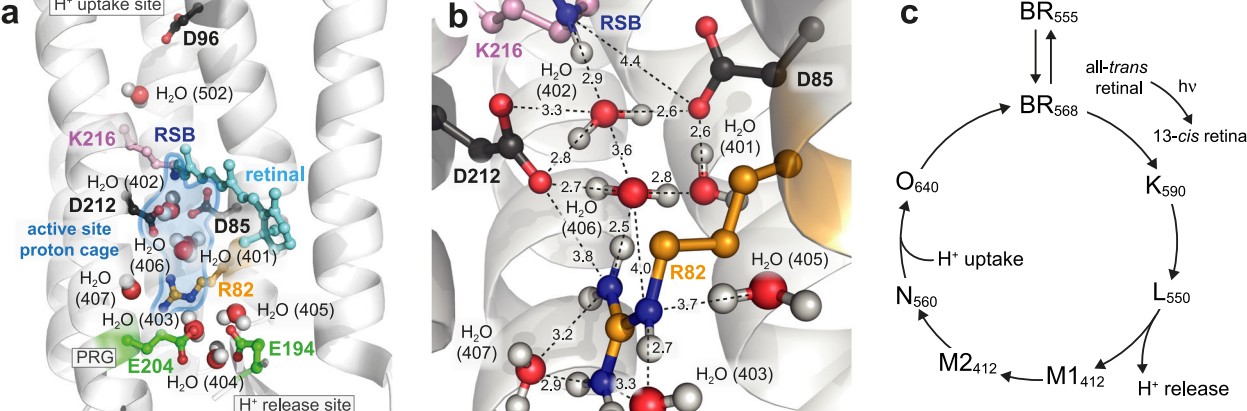

**Fig. 1 The light-driven proton pump bacteriorhodopsin (BR, pdb 1c3w). a** The proton transport pathway of BR involves D96 at the proton uptake site, D85, the retinal Schiff base (RSB) that is covalently bound to K216, and R82 in an active site proton cage. The extracellular proton release site is composed of the proton release group (PRG) with two glutamic acids (E194 and E204) and three water molecules. **b** The active site proton cage includes the RSB, D85, R82, and the $H_2O$ molecules 401, 402, and 406. Dashed lines indicate distances in Å. **c** The photocycle of BR. The numbers correspond to the maximum absorption wavelength of the respective intermediate.

distribution between a $H_2O$ molecule and two carboxyl groups, a simplified model for proton displacement in the D85 environment.

## Results

**Resonance assignments of exchangeable protons in BR**. As a first step, we monitored the efficiency of proton back-exchange by recording a proton-detected, two-dimensional $^{15}N$–$^{1}H$ correlation using cross polarization-based magnetization transfers exploiting dipolar, through-space couplings on light-exposed purple membranes with fast MAS (Supplementary Fig. 1)[26,27]. The spectrum indicates significant back-exchange of approximately 70 amino acids as shown by the respective backbone amide signals, corresponding to 28% of the residues. Based on a set of triple-resonance experiments[28], we could assign approximately 59% of these signals (Supplementary Figs. 1, 2 and Supplementary Table 1), reflecting indeed amide groups of residues in loops or in solvent accessible areas as is often seen in MAS NMR studies of deuterated membrane proteins (Supplementary Fig. 3)[29]. The well-dispersed amide group signals with a chemical shift pattern characteristic for α-helices clearly suggest that BR adopts its native seven-transmembrane helix fold (Supplementary Fig. 1). The $^{15}N$–$^{1}H$ correlation and the observed $^{13}C\alpha$ and $^{13}C\beta$ chemical shifts agree well with previous NMR studies of purple membranes[30,31]. However, none of these assigned backbone resonances correspond to residues involved in proton pumping; we therefore applied the following strategy to study sites that contribute to proton displacements during BR molecular action.

Protons in the proximity to the carboxyl groups of D96, D85, at the guanidinium group of R82, and involving nearby $H_2O$ molecules were probed by a combination of two-dimensional $^{15}N$–$^{1}H$ (Fig. 2a, b), $^{13}C$–$^{1}H$ (Fig. 2a, c), both cross polarization-based, and $^{1}H$–$^{1}H$ exchange spectroscopy (Fig. 2a, d) of BR wildtype purple membranes. We further investigated in this way BR mutated at the proton donor of the RSB (D96), the acceptor of the RSB proton (D85) and the proton shuttle R82: D96N, in which the reprotonation of the RSB is dramatically slowed down[32], D85T that features a red-shift of the BR-absorption and complete inactivation of proton pumping[33,34], and R82Q, in which BR-absorption is red-shifted, the RSB-p$K_a$ lowered, and proton shuttling from D85 to the proton release group severely hampered[35,36].

Comparison of the $^{13}C$–$^{1}H$ correlations of wildtype purple membrane and of the mutant D96N yields the C$^{\gamma}$ chemical shift of D96, in accordance with earlier assignments by the Engelhard group[37], and the shift of the attached proton (Fig. 2c and Supplementary Table 2). The $^{1}H$–$^{1}H$ exchange spectrum of D96N confirms the assignment of the signal at 11.0 ppm $^{1}H$ chemical shift to D96–H$^{\delta2}$ (Fig. 2d and Supplementary Fig. 4). In the BR wildtype spectrum, the carboxylic acid proton of D96 shows a cross peak at the water frequency due to chemical exchange with $H_2O$ (Fig. 2a, d). Employing the cross polarization-based $^{13}C$–$^{1}H$ spectrum of the D85T mutant (Fig. 2c), the signal at 12.1 ppm $^{1}H$ chemical shift in the wildtype spectra can be formally assigned to D85–H$^{\delta2}$ (Fig. 2a, c, d and Supplementary Table 2). The observed D85–C$^{\gamma}$ resonance agrees well with the assignment obtained in earlier studies[37]. This is also confirmed by the missing diagonal and cross peaks at the D85–H$^{\delta2}$ frequency in the $^{1}H$–$^{1}H$ exchange spectrum of D85T, even when taking different signal-to-noise ratios into consideration (Fig. 2d and Supplementary Fig. 4). The H$^{\eta}$ of R82 can be assigned to a signal at 6.2 ppm $^{1}H$ chemical shift as it disappears in the $^{15}N$–$^{1}H$ spectrum of the BR mutant R82Q (Fig. 2b and Supplementary Table 2).

**Proton exchange detected at room temperature by MAS NMR**. The observed proton signals, including their cross peaks in the exchange spectra are intriguing. In general, the presented exchange spectra include contributions from chemical and spin exchange, however, we assume the latter to be of minor importance due to the distances between the diluted protons investigated and the high MAS frequencies applied, which significantly reduce spin diffusion based on dipolar couplings. The signal at 12.1 ppm is assigned to be in the vicinity of the D85 side chain carboxyl group and may reflect either protonation of D85 or a hydronium ion (occurring typically at 12–14 ppm[38]), or the chemical shift average of both. Formally, several tautomeric states are possible for the cluster involving the RSB, water 402, and D85 (Fig. 2e). For water 402 that is close to the carboxyl group of D85 (Fig. 1b), proton displacement can lead to partially populated hydroxide and hydronium ion states as well as to a protonated D85 form (Fig. 2e). The hydroxide ion occurs in conjunction with a D85 carboxyl group proton, and the hydronium ion requires D85 to be non-protonated (Fig. 2e). D85 should therefore be considered as partially ionized under the applied experimental conditions. We will refer to the respective proton signal as D85–H$^{\delta2}$. To analyze this situation further, cross polarization build-up experiments were additionally recorded for the D85 and D96 C$^{\gamma}$–H$^{\delta2}$ cross peaks (Supplementary Fig. 5). The D96 signal builds up faster than the D85 peak, indicating that the carboxylic acid proton in the D85 case is, on average, more distant from the C$^{\gamma}$ and therefore closer to $H_2O$ in comparison to D96. However, varying D85 and D96 side chain mobilities represent an additional source for the observed differences in cross polarization build-up. Similar to D96–H$^{\delta2}$, D85–H$^{\delta2}$ exhibits a cross peak to $H_2O$ in the BR wildtype exchange spectrum (Fig. 2a, d). In contrast to D96–H$^{\delta2}$, the exchange peak of D85–H$^{\delta2}$ at the water frequency has a much higher intensity as the corresponding diagonal peak. Intensities of diagonal peaks in exchange spectra are directly correlated to the populations, whereas those of cross peaks are additionally modulated by the kinetics of the exchange process between the two nuclei giving rise to the cross peak as well as by the populations of their respective states. In line with the above-mentioned cross polarization build-up experiments (Supplementary Fig. 5), the observed intensity differences therefore suggest that the localization of D85–H$^{\delta2}$ is in favor of the chemical environment of water, which could explain that it was not observed by vibrational spectroscopy so far. Collectively, our data provide evidence that D85–H$^{\delta2}$ is exchanged within the hydrogen bond between D85 and one of the $H_2O$ molecules close to it (Figs. 1b, 2a, c–e and Supplementary Figs. 4 and 5).

The BR wildtype exchange spectrum shows a cross peak between R82 and D85–H$^{\delta2}$ at 6.2 and 12.1 ppm $^{1}H$ chemical shift, respectively, which may reflect a signal due to long-range proton exchange between the carboxyl group of D85 and the guanidinium group of R82, potentially via the structurally close water molecules 401 and 406 (Figs. 1b, 2a, d and Supplementary Fig. 4)[5]. In general, we cannot distinguish different water populations spectroscopically, including bulk water and protein-bound water molecules. Thus, proton exchange between amino acids and the sum of all water molecules is observed. This may be resolved if smaller $^{1}H$ linewidths could be achieved by even higher MAS frequencies, or through $^{17}O$ spectroscopy at very high magnetic fields. As expected, the R82–H$^{\eta}$ signal is missing in the $^{1}H$–$^{1}H$ exchange spectra of the R82Q and D85T mutants, since R82–H$^{\eta}$ is indirectly observed via D85–H$^{\delta2}$ detection in the wildtype (Fig. 2a, d and Supplementary Fig. 4). For BR wildtype, these findings show that there is a proton translocation pathway to the R82 guanidinium group and, consequently, R82 should be considered as an intermediate in proton relocation serving as an

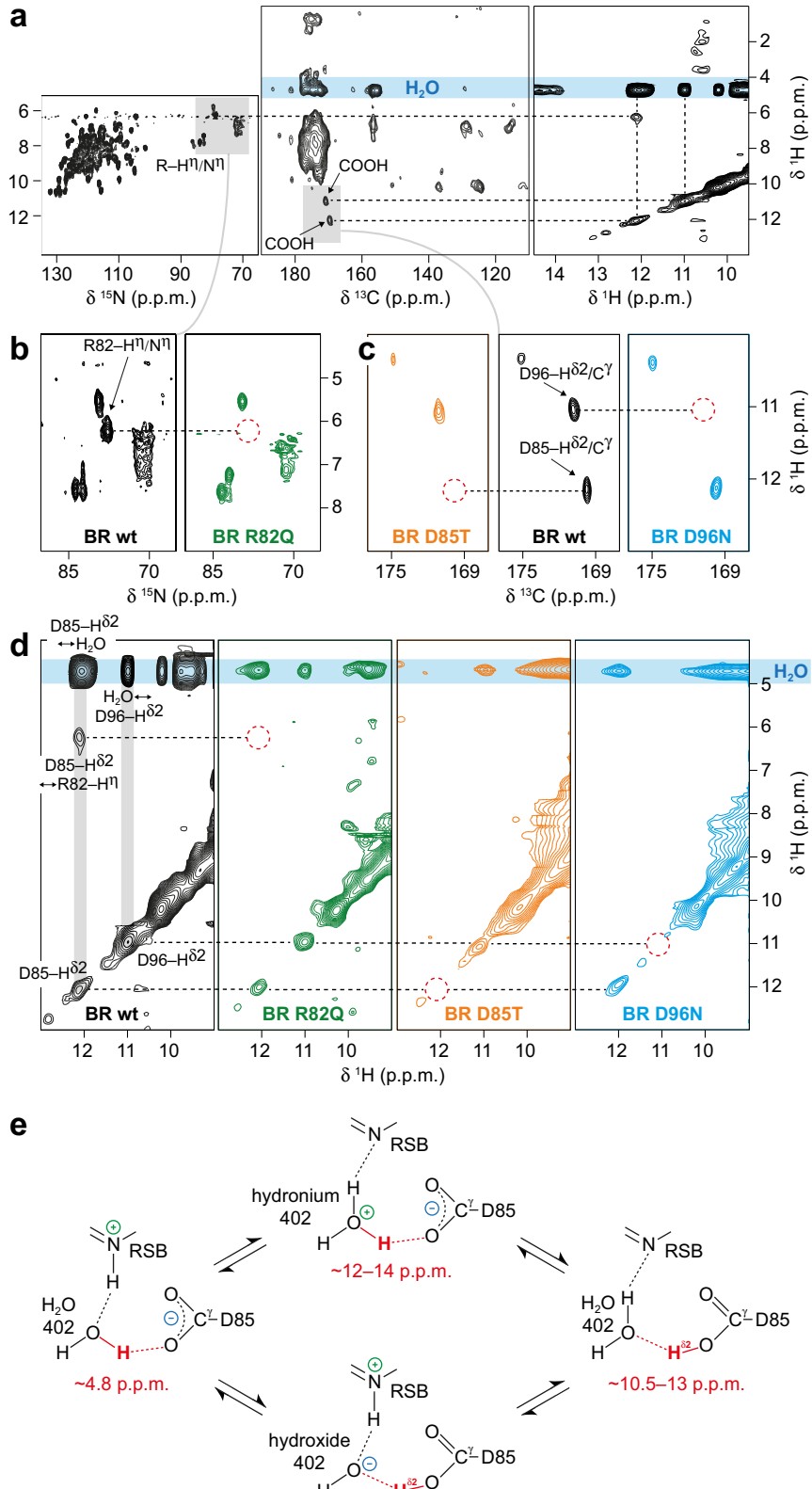

**Fig. 2 Protonation states and chemical exchange of protons at R82, D96, D85, and water molecules in bacteriorhodopsin (BR) probed by proton-detected MAS NMR at room temperature. a** $^{15}$N–$^{1}$H (left), $^{13}$C–$^{1}$H (middle), and $^{1}$H–$^{1}$H exchange (right) spectra of BR wildtype. **b** $^{15}$N–$^{1}$H correlations of BR wildtype (wt, black spectrum) and the mutant R82Q (green spectrum). **c** $^{13}$C–$^{1}$H correlations of BR wildtype (wt, black spectrum) and the D85T and D96N mutants (orange and blue spectra, respectively). The mutants are used to assign the signals of D96–H$^{δ2}$ (11.0 ppm), D85–H$^{δ2}$ (12.1 ppm) and R82–H$^{η}$ (6.2 ppm). Red dashed circles indicate missing signals resulting from mutations that allow the assignment. **d** $^{1}$H–$^{1}$H exchange spectra of BR wildtype (wt, black spectrum), R82Q (green spectrum), D85T (orange spectrum) and D96N (blue spectrum). **e** At $H_2O$ 402, both hydronium and hydroxide ions are possible, affecting the proton localization through exchange within the hydrogen bonds to the retinal Schiff base (RSB) and D85. The observed chemical shifts for the proton highlighted in red agree with tautomeric structures involving $H_2O$ 402, a hydronium 402 and a carboxyl group proton at D85.

active proton donor and acceptor on the way from D85 towards the proton release group.

In summary, an exchange signal is observed between D85 and R82, indicating a proton pathway between these two residues. In addition, both the D85 and D96 carboxyl groups show chemical exchange of protons with $H_2O$ in the BR dark-state upon illumination as seen after $^1H/^2H$ back-exchange. Illumination allowed protons to enter the pore of BR via exchangeable sites involved in the proton transport pathway. We conclude that in both cases, D85 and D96, a proton undergoes displacement between the carboxyl group and $H_2O$. The exchange between D85 and $H_2O$ may lead to a hydronium ion state at $H_2O$ 402.

**RSB proton exchange observed by MAS NMR at variable temperatures**. In agreement with the occurrence of a dynamic process (Fig. 2e), we see exchange broadening of the RSB proton or nitrogen signal (Fig. 3a), indicating that the so far observed exchange in the active site proton cage involves the RSB NH moiety as well (Figs. 1b, 2e). In a previous study, the RSB nitrogen signal could be observed at around 100 K in the spectral region of 165–175 ppm for the BR dark-state[10]. In a similar, recent experimental approach involving measurements at around 100 K, the signal of the RSB proton itself could be detected at 13.2 and 12.2 ppm for BR$_{555}$ (13-*cis*,15-*syn* retinal configuration) and BR$_{568}$ (13-*trans*,15-*anti* retinal configuration), respectively[39]. At room temperature, such proton resonances are not observed. To examine the situation further, we performed proton-detected MAS NMR measurements between 97 and 273 K (Fig. 3a). At temperatures below 200 K, we observe well-resolved signals of the RSB NH moiety at 13.2 ppm (BR$_{555}$, $^{15}N$ chemical shift of 173.5 ppm) and 12.2 ppm $^1H$ chemical shift (BR$_{568}$, $^{15}N$ chemical shift of 165.4 ppm) in agreement with the previous studies (Fig. 3a and Supplementary Table 2)[10,39]. Above 200 K, however, the signal-to-noise ratios of the two RSB proton resonances decrease and both signals vanish towards 273 K, where they are not observable anymore (Fig. 3a, red spectrum in the very right panel, bottom). In general, multiple factors such as temperature-induced changes in cross polarization efficiency and Boltzmann distribution can lead to different sensitivity in these experiments. To compensate for this, we recorded the spectra with optimized cross polarization transfers and similar measurement time by adjusting the number of scans, and taking into account that $T_1(^1H)$ changes with temperature, applying different recycle delays (in each experiment set to $1.3 \times T_1(^1H)$). Another possible explanation for the disappearance of the two RSB proton signals could be conformational exchange between the 13-*cis*,15-*syn* and 13-*trans*,15-*anti* retinal configurations in the µs/ms time regime. If this would be the case, however, it would not take 21 min to reach equilibrium of the 40/60% distribution of these isomers as measured by Oesterhelt et al.[40]. The low B-factors of the active site[5], including the RSB, in the crystal structure and the well-defined, sharp NMR signals observed at low temperature further indicate high structural homogeneity. This excludes other conformational dynamics in the protein that could potentially lead to the observed line broadening, in agreement with previous studies employing molecular dynamics simulations by the Elstner group[41]. We thus conclude that the RSB proton signal loss is attributed predominantly to line broadening caused by chemical shift exchange. Such an effect can result from proton exchange leading to disappearance of signals in NMR spectra due to coalescence. Therefore, these data suggest that the RSB proton is displaced at room temperature, involving exchange with $H_2O$ 402 (Figs. 1b, 2e). Remarkably, the $^1H$ chemical shifts of 12.2 and 13.2 ppm for the RSB proton agree again well with a signal expected for a hydronium ion, similar to the situation of D85 (Fig. 2e). This

exchange may thus be coupled to the observed proton displacement between D85 and $H_2O$ 402, causing potential collective proton exchanges within the hydrogen bonding networks (Fig. 2e). Even though a high proton affinity of the RSB has been measured by bulk phase pH titration[42], so far no evidence exists for a proton path to the exterior of the protein in the dark-state and the reported p$K_a$ of 13.3 must be therefore considered as apparent, still enabling proton oscillation at the RSB and water 402. After long discussions in the literature, it has been accepted that D85 and D212 are the only deprotonated aspartates in darkness[32,43,44]. However, our findings suggest that D85 is exchanging rapidly a proton with $H_2O$ 402 and/or 401 even in the dark, presumably eased by or coupled to proton displacement between the RSB and $H_2O$ 402.

**Characterization of proton exchange between carboxyl groups and $H_2O$**. To provide a model for proton displacement in a protein environment, we performed ab initio molecular dynamics simulations to compute the proton distribution in a simplified configuration, consisting of an $H_2O$ molecule located between two carboxyl groups with one excess proton, i.e., the system has a net charge of 1 $e^-$ (Fig. 3b, see Methods section for details). This model mimics the water molecule 402 that is coordinated by D212 and D85 (Fig. 1b) and thus enables an analysis of primary, thermally activated proton oscillations in the BR active site. The distances between the two carboxyl groups, and the positions of the central water molecule and of the protons are not fixed but are subject to thermal fluctuations. Depending on the distance between the carboxylic acid and water oxygens ($r_{O_CO_W}$), the relative position of proton 2, described by an asymmetry coordinate ($r_{O_CH}-r_{O_WH}$, Fig. 3b), exhibits different behavior[45]. Note that a negative asymmetry refers to the proton localized near the carboxyl group oxygen, and a positive asymmetry near the water oxygen. For $r_{O_CO_W}$ of 2.4–2.6 Å, intermediate positions of proton 2 are frequently observed, while for large values of $r_{O_CO_W}$ the proton tends to be localized near the central water molecule, as can be seen from the free energy distribution in Fig. 3c. This means that dynamic proton exchange is observed in time-resolved trajectories, from which a mean residence time of about 1 ps is determined (Fig. 3d). The analysis of the typical distances between the protons and oxygen atoms reveals that bifurcated, three-centered hydrogen bonds[46] do not appear in the model system (Supplementary Fig. 6). The proton transfer from the water oxygen to a carboxyl group involves a free-energetic barrier of about 3 $k_BT$ as a function of the asymmetry reaction coordinate $r_{O_CH}-r_{O_WH}$, as shown in Fig. 3e. In fact, Eckert and Zundel found a 6 $k_BT$ barrier for a fixed $r_{O_CO_W}$ of 2.65 Å from static ab initio self-consistent field calculations[47]. It is not surprising that the barrier height is smaller in our dynamic simulation which allows for kinetic relaxation of all positions. In BR, $r_{O_CO_W}$ is 2.6 Å between D85 and the well-defined $H_2O$ 402 that shows low B-factors in the crystal structure solved at cryogenic temperatures (100 K) (Fig. 1b)[5]. However, the NMR experiments were performed at around 290 K, therefore a slightly shorter distance $r_{O_CO_W}$ for D85 and $H_2O$ 402 is possible as well. Such variation in distances agrees well with molecular dynamics simulations of the Gerwert group[48], and has been observed by time-resolved serial femtosecond crystallography at room temperature[2,3]. In our experiments, a proton undergoing similar displacement gives rise to an NMR signal at 12.1 ppm, reflecting the chemical shift of D85–H$^{\delta2}$ or its time-weighted average with the proton chemical shift of a hydronium ion (12–14 ppm) (Fig. 2a, c–e and Supplementary Figs. 4 and 5). Worth of note, both $H_2O$ 401 and 402 have the same distance of 2.6 Å to D85–O$^{\delta2}$ and form a pentameric hydrogen bond arrangement together with D85, D212, and $H_2O$ 406 (Fig. 1b)[4,5].

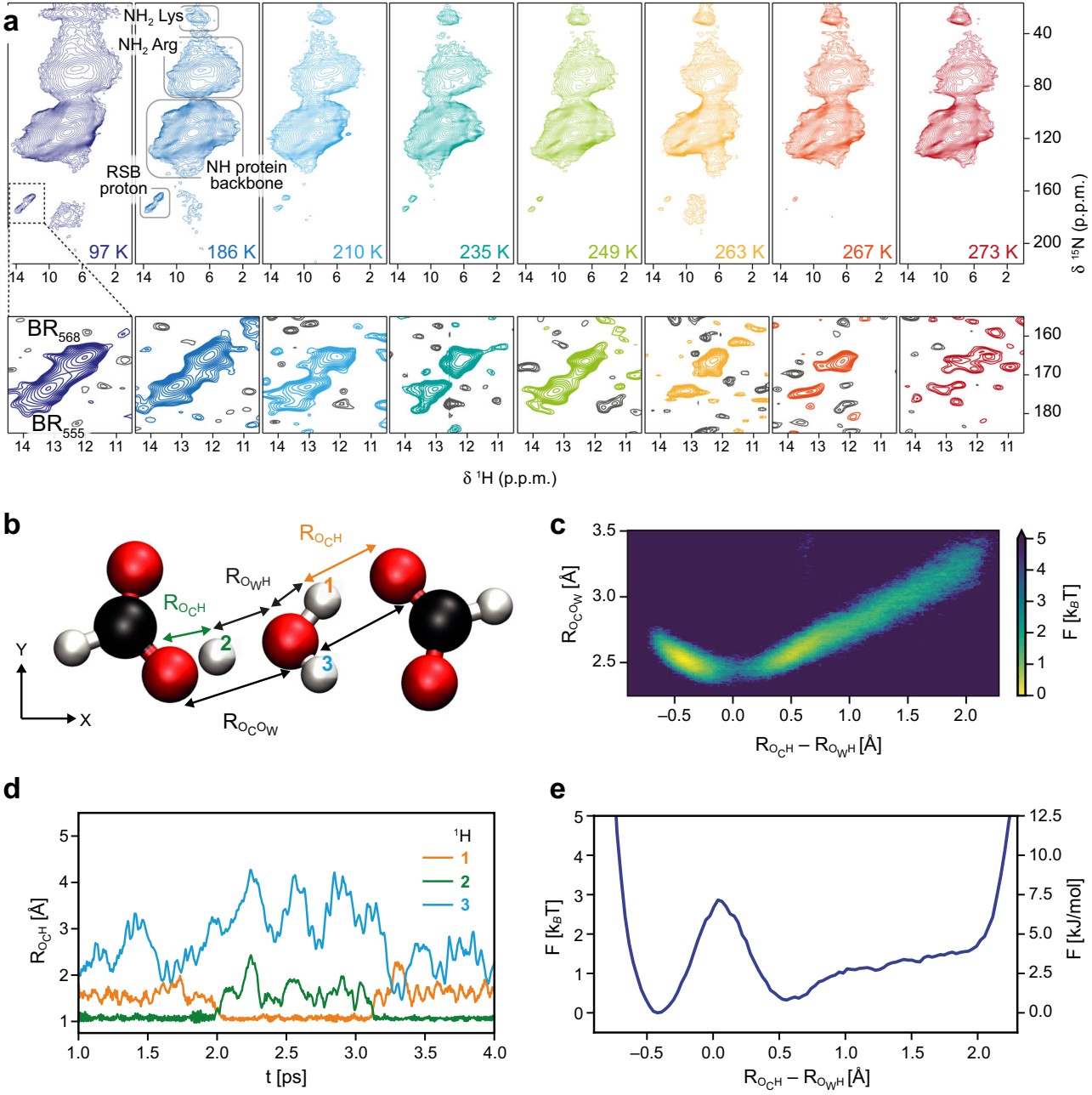

**Fig. 3 Observation of retinal Schiff base (RSB) proton dynamics by MAS NMR and characterization of proton exchange by *ab initio* molecular dynamics simulations in a model system. a** Proton-detected, two-dimensional $^{15}N$–$^{1}H$ correlation spectra of purple membrane in bacteriorhodopsin (BR) dark-state recorded at temperatures ranging from 97 K (dark blue spectrum) to 273 K (red spectrum). The spectral region of the RSB proton signals for $BR_{568}$ and $BR_{555}$ is plotted at the noise level (bottom panels, positive contours are color-coded according to the temperature and negative contours are shown in gray). **b** The proton distribution is analyzed by ab initio molecular dynamics simulations in a model system consisting of one $H_2O$ molecule, two carboxyl groups and one excess proton which are all subject to thermal positional fluctuations. The distance between carboxyl group and water oxygen atoms ($R_{O_CO_W}$) and the excess proton's relative asymmetry with respect to the two oxygens ($R_{O_CH}$–$R_{O_WH}$) are used as effective reaction coordinates. **c** The free energy landscape of the proton is shown as a function of the coordinates, as defined in **b**. **d** Trajectories of the distances, $R_{O_CH}$, of the three central hydrogens labeled as in **b** to the respective closest carboxyl group oxygen. The proton exchange is well visible as a fast jump process: proton 1 resides near the closest carboxyl group oxygen for about 1 ps in the time interval from $t = 2.0$ ps to $t = 3.1$ ps. **e** The free energy of the protons projected onto the asymmetry coordinate ($R_{O_CH}$–$R_{O_WH}$) indicates a low proton transfer barrier of about 3 $k_BT$.

## Discussion

In the present study, we characterize protonation dynamics in the BR dark-adapted state, revealing a proton cage close to the Schiff base and displacement of protons at D85 and H2O 402. The proton cage extends to R82 which is involved in a hydrogen bonding network with water molecules and the carboxyl group of D85 (Fig. 1b)[4,5]. R82 acts as a regulator of the D85 p$K_a$, but a

transient deprotonation of it has neither been unequivocally observed nor excluded[19,49,50]. The detected protonated form of R82 and the exchange signal correlating R82–H$^\eta$ and D85–H$^{\delta2}$ suggest chemical exchange between both residues within this network, presumably via the water molecules 401 and 406 as these, based on the dark-state crystal structure solved at 100 K, may provide a direct connection via a hydrogen bond network

(Figs. 1b, 2a, b, d)[5]. This reveals a proton path between D85 and R82, highlighting the involvement of the R82 guanidinium group in proton pumping. The observed proton exchange between the RSB, water molecules, D85 and R82 indicate the existence of a pathway across the proton cage at the BR active site (Figs. 1a, b, 2, 3a). After retinal isomerization upon light reception, the underlying exchange equilibrium involving R82 and D85 might be shifted towards the protonated form of D85 during the $L_{550} \rightarrow M1_{412}$ transition. Potentially, this change in equilibrium is facilitated by the structural changes detected by time-resolved serial femtosecond crystallography[2,3]. The MAS NMR exchange spectra show exchange of R82–H$^{\delta 2}$ already in the dark-state which supports the possibility of R82 deprotonation in the $L_{550} \rightarrow M1_{412}$ transition (Figs. 1b, c, 2a, d).

The detected protonation of D96 in the BR dark-state agrees well with previously obtained Fourier-transform infrared spectroscopy results[4]. D96 serves as the central binding position in the proton uptake site in conjunction with $H_2O$ molecules located at the cytoplasmic entry of the protein pore (Fig. 1a). It has been proposed that the D96–H$^{\delta 2}$ is relocated via a transiently established chain of three water molecules to the RSB in a Grotthuss-type mechanism upon the $M2_{412} \rightarrow N_{520}$ transition as revealed by molecular dynamics simulations (Fig. 1c)[18]. The observed proton exchange between D96 and water in $BR_{568}$, however, requires $H_2O$ molecules close to the carboxyl group already in the BR dark-state. The next water molecule, $H_2O$ 502, occurs at a distance of 5.0 Å to the D96–O$^{\delta 2}$ along the proton transport pathway towards the RSB, as measured in the crystal structure solved at 100 K (Supplementary Fig. 7)[5]. We thus suggest that, in the BR dark-state, water molecule 502 may be rather dynamic at room-temperature in contrast to the situation at cryogenic temperatures. The possibility of proton displacement between D96 and water molecules at the cytoplasmic entry site of the BR pore can be largely excluded due to low water accessibility in the dark-state[48]. Additional water molecules not resolved in the crystal structures represent another possible explanation for the proton exchange observed between these sites. In any case, the D96–H$^{\delta 2}$ can exchange with a water molecule in the dark-adapted $BR_{568}$ state.

In conclusion, we have directly observed signals of exchangeable protons at key sites including the RSB, D96, D85, and R82 in purple membranes using proton detection by fast MAS NMR. We have further detected chemical exchange of these protons with water, in particular within the active site proton cage of dark-adapted $BR_{568}$. Our experiments point towards oscillations between intermittently occurring tautomeric structures including a hydronium ion in place of water 402 and a protonated D85 form. The ab initio molecular dynamics simulations show proton residence times of 1 ps in a model system. Displacement of the RSB proton corroborates coupling of the observed proton exchange processes between the RSB, D85, and R82 via $H_2O$ molecules in the active site proton cage, indicating reversible proton displacement in the dark-state. Intriguingly, the distances between the RSB and D85 hydrogen bond acceptors and the oxygen of water 402 are 2.6 Å. This is slightly longer than expected for an ideal hydrogen bond. We assume this promotes the oscillation between different states, as highlighted by the ab initio molecular dynamics simulations on proton displacement between $H_2O$ and two carboxyl groups which are reflecting the counter ion geometry. The potential involvement of R82 in proton transport is supported by an exchange signal involving the carboxyl group of D85 in dark-adapted BR. As proton pumping by BR is a directed process, we suggest that retinal isomerization makes the observed chemical exchange within the proton cage irreversible through disrupting the proton oscillation via tilting of the RSB, resulting in the extracellular release of a proton.

## Methods

**Preparation of $^2H,^{13}C,^{15}N$-labeled purple membrane.** For production of purple membranes of bacteriorhodopsin, the *Halobacterium salinarum* strains BR-wild-type, BR-R82Q, BR-D85T, and BR-D96N from Janos Lanyi, Dieter Oesterhelt and Markus Lange, Actilor GmbH were used. The following procedures were the same for all four strains. Cells were grown in deuterated celtone medium for at least 10 days at 37 °C at 100 rpm in shaking flasks, after pre-culturing in protonated medium. In 200 mL, the sterile filtered medium contained, both in deuterated and protonated form: 1 g $^{13}C,^{15}N$-celtone, 50 g NaCl, 0.4 g KCl, 4 g MgSO$_4$, 0.04 g CaCl$_2$, 0.6 g Na-citrate ($C_6H_5Na_3O_7*2H_2O$), 0.4 mg Biotin, 0.4 mg Thiamin, 0.06 µg MnSO$_4$, 0.72 µg FeCl$_2$, 0.088 µg ZnSO$_4$, and 0.01 µg CuSO$_4$ at pH 7. Cells were harvested at 7000 rpm for 20 min at 4 °C in a Beckman JLA 8.1 rotor with 1 L tubes. The supernatant was discarded and cells were resuspended in 6 mL basal salt (250 g/L NaCl, 20 g/L MgSO$_4$*7H$_2$O and 2 g/L KCl at pH 7). After adding DNAse (grade II) and 65 mL H$_2$O, the cell suspension was dialyzed over night at 4 °C against H$_2$O under stirring. Membranes were then collected via centrifugation for 60 min at 40,000 rpm and 4 °C with a Sorvall 90SE ultracentrifuge using a 45Ti rotor. 3 mL of 50 mM Tris-HCl at pH 7.4 were used to resuspend the pellet, and the membranes were homogenized and fractionated by density gradient centrifugation using saccharose (30–70%) for 14 h at 22,000 rpm and 15 °C with a Sorvall 90SE ultracentrifuge using a TST-28 rotor. The purple membrane fraction was collected and concentrated for 1 h at 40,000 rpm and 4 °C with a Sorvall 90SE ultracentrifuge using a 45Ti rotor. After washing with 50 mM Tris-HCl at pH 7.4, followed by homogenization and concentrating again (this procedure was performed twice), the purple membranes were stored at −20 °C.

**Sample preparation for solid-state MAS NMR measurements.** The purple membranes were diluted to 0.01 OD in 90%/10% $^1H_2O/^2H_2O$ 50 mM Tris-HCl at pH 7.4 and illuminated (a 595 nm filter was used) for 4 h under stirring in a water-cooled cuvette at 15 °C with a home-build illumination set-up. We thus assume that each bacteriorhodopsin molecule went through the photocycle during this photo-equilibration, thereby pumping protons and back-exchanging protons at key sites of the proton transport pathway (including the D85, D96, and R82 side chains, the retinal Schiff base and water molecules). The purple membranes were then collected by ultracentrifugation for 2 h at 150,000 × g and 4 °C, and packed into 1.9 mm (for 20 and 40 kHz MAS experiments) or 1.3 mm (for 60 kHz MAS experiments) Bruker MAS NMR rotors using home-made filling tools. Rotors were sealed with silicone rubber disks to avoid loss of liquid during MAS. Before starting acquisition of MAS NMR experiments, the samples were equilibrated in the dark (in the MAS NMR probe inside the magnet) for at least 1.5 h to allow for establishing the BR dark-adapted state, i.e., the mixture of the 13-*cis*,15-*syn* and 13-*trans*,15-*anti* retinal configurations, which has been measured to reach equilibrium after 21 min[40].

**Solid-state MAS NMR measurements.** NMR experiments were performed on Bruker Avance III spectrometers with $^1H$ Larmor frequencies of 800 and 900 MHz using 1.3 mm triple-resonance (HCN) and 1.9 mm four-channel (HCND) Bruker MAS probes, respectively. The variable temperature was controlled using a Bruker cooling unit and adjusted to 260 K and 230 K for 60 kHz MAS experiments with the 1.3 mm probe and 40 kHz MAS experiments with the 1.9 mm probe, respectively. According to external temperature calibration, this corresponds to actual sample temperatures of 291 K for the 1.3 mm probe and 293 K for the 1.9 mm probe. The temperature-dependent detection of RSB proton dynamics was performed at 20 kHz MAS on a 800 MHz Bruker Avance III spectrometer using a 1.9 mm four-channel dynamic nuclear polarization probe. A cross polarization-based (H)NH pulse sequence with MISSISSIPPI solvent suppression[51] was applied at actual sample temperatures of 97, 186, 210, 235, 249, 263, 267, and 273 K. The temperature was adjusted with a Bruker LT/MAS unit. The $^1H$–$^{15}N$ and $^{15}N$–$^1H$ transfers were optimized around $w_1^H = 3\omega_R/2$ and $w_1^X = 1\omega_R/2$. For each experiment, the recycle delay was set to $1.3 \times {}^1H(T_1)$ as measured via inversion recovery. The number of scans was adjusted such that each experiment had the same measurement time of about 16 h with identical acquisition times and spectral windows.

The $^1H$–$^1H$ exchange spectra were recorded at 40 and 60 kHz MAS at sample temperatures of 293 and 291 K, respectively, using a conventional z-exchange scheme with a mixing time of 10 ms. For water suppression, 1–1 hard pulses were applied just before detection[52]. Both pulses were adjusted to 90°, separated by a delay of 45 µs corresponding to $1/4\nu_{max}$ (with $\nu_{max}$ being the difference in frequency of the protons of interest and the water protons).

In all other NMR experiments, cross polarization for heteronuclear magnetization transfers and MISSISSIPPI solvent suppression (except in $^1H$–$^1H$ exchange experiments) was used. (H)NH, (H)CANH, (H)CA(CO)NH, (H)CONH, (H)CO(CA)NH, (H)CBCANH, and (H)CBCA(CO)NH spectra were recorded with pulse sequences according to Barbet-Massin et al.[28] and experimental parameters as used in Nieuwkoop et al.[53]. Tangential shapes were applied for heteronuclear cross polarization transfers. The $^1H$–$^{15}N$ and $^1H$–$^{13}C$ cross polarization steps were optimized around $w_1^H = 3\omega_R/2$ and $w_1^X = 1\omega_R/2$. The $^{13}C$–$^{15}N$ specific cross polarization transfers were optimized between $w_1^C = 3\omega_R/4$ and $w_1^C = 5\omega_R/4$, and $w_1^N = 1\omega_R/4$. For homonuclear ($^{13}C$–$^{13}C$) transfers, DREAM and INEPT was used

at 40 kHz and 60 kHz MAS, respectively. WALTZ-16 was applied for $^{1}$H, $^{13}$C, and $^{15}$N decoupling during indirect evolution periods.

The $^{1}$H–$^{13}$C correlation spectra were recorded at 40 kHz (293 K sample temperature) and 60 kHz (291 K sample temperature) MAS either with $^{13}$C-detection using one cross polarization transfer or with the proton-detected (H)CH pulse sequence employing two cross polarization steps. The latter was acquired again with MISSISSIPPI solvent suppression. For both $^{1}$H–$^{13}$C cross polarization and $^{13}$C–$^{1}$H cross polarization, the applied amplitudes were optimized around $w_1^H = 3\omega_R/2$ and $w_1^X = 1\omega_R/2$.

All spectra have been processed with Bruker Topspin Versions 3.4 or 4.0 and analyzed with CcpNmr Analysis Version 2.4. Gaussian (typically with a maximum at 0.04 and 40 Hz line broadening, except for the variable temperature series spectra for which a maximum at 0.1 and 100 Hz line broadening was used) and sine squared (sine bell shift of 2) apodization functions have been applied in the direct and indirect dimensions, respectively. For all spectra, baseline correction with a polynomial of degree 5 was used.

**Ab initio molecular dynamics simulations**. In order to quantify the excess proton localization in the vicinity of a carboxyl group, ab initio molecular dynamics simulations were performed in a model system, using a TZV2P basis set, the BLYP exchange-correlation functional and D3 dispersion correction in the CP2K 2.4 environment[54–57]. The model system consists of a pair of deprotonated carboxyl groups, as commonly assumed in BR at D212 and D85, with a water molecule and one excess proton in between as shown in Fig. 3b. Note that this model agrees well with the BR crystal structure shown in Fig. 1b, which resolved a water molecule located right in between the D85 and D212 side chains. The simulation box size was optimized to be $15 \times 8 \times 8$ Å$^3$. The molecules were kept near the central axis of the simulation box by constraining the carbon atoms in $y$ and $z$ direction using Lagrangian multipliers implemented in the Shake algorithm. In addition, the water oxygen atom is constraint by a quadratic potential of 50 meV/Å$^2$ in $y$ and $z$ direction to keep it close to the central axis. No constraints are applied along the axis connecting the two carbon atoms. A 40 ps molecular dynamics simulation was performed under NVT conditions at 300 K by coupling all atoms to a Nosé-Hover chain of size three and decay time constant of 100 fs[58]. The first 5 ps were dismissed for equilibration. Consequently, nine independent molecular dynamics simulations (5 ps each) were performed under NVE conditions starting from statistically independent snapshots of the NVT data. Proton distributions are computed from both, NVT and NVE simulations, thus from a total simulation time of 80 ps.

**Reporting summary**. Further information on research design is available in the Nature Research Reporting Summary linked to this article.

## Data availability
The datasets generated during and/or analyzed during the current study are available from the corresponding author on reasonable request. The obtained NMR chemical shifts are deposited in the Biological Magnetic Resonance Bank database (accession number 50085).

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

## Acknowledgements

We thank Margrit Michalsky and Thi Bich Thao Nguyen for excellent technical assistance and production of purple membranes, and Janos Lanyi, Dieter Oesterhelt and Markus Lange, Actilor GmbH, for the *Halobacterium salinarum* strains BR-R82Q, BR-D85T, and BR-D96N. Initial efforts in experimental design by Marcella Orwick-Rydmark and support in NMR data analysis by Lisa Gerland is kindly acknowledged. This study received funding by the Deutsche Forschungsgemeinschaft through SFB 1078 to Roland R. Netz (C1), Peter Hegemann (B1, B2), and Hartmut Oschkinat (B1). Daniel Friedrich received support by the Human Frontier Science Program (HFSP, LT000022/2019-L) and is a non-stipendiary Fellow of the European Molecular Biology Organization (EMBO, ALTF 35-2019).

## Author contributions

D.F., P.H., and H.O. designed the study. D.F. and A.J.N. designed and performed NMR experiments. F.N.B. and R.R.N. designed, performed and analyzed ab initio molecular dynamics simulations. D.F. and H.O. analyzed and interpreted data. D.F., P.H., and H.O. wrote the paper with contributions from all authors.

## Competing interests

The authors declare no competing interests.
