## [Peer Review File · Communications Biology]

Reviewers' comments:

Reviewer #1 (Remarks to the Author):

This paper describes solid state NMR measurements on bacteriorhodopsin, designed to provide information about movements of protons within the bR proton-pumping channel that are relevant to bR function. The authors prepare triply-labeled (2H,15N,13C) purple membrane samples and allow limited back exchange of 2H to 1H, thus limiting the number of 1H solid state NMR signals and leading to relatively simple spectra in which certain crucial 1H signals can be resolved and assigned.

In principle, measurements of this type could provide important new information. However, I am not sure that the authors have succeeded in obtaining definitive new information, at least in part because the authors' description of their results is unclear. My overall evaluation is that this work is interesting and potentially important, but the authors need to state their conclusions more clearly and explain the experimental bases for these conclusions more succinctly.

The authors should address the following points:

1. The description of the solid state NMR experiments begins with assignments of backbone 1H, 15N, and 13C signals from triple-resonance experiments. It seems that none of residues with assigned backbone signals participate in proton pumping. Is this true? If so, the authors should clarify that these triple-resonance experiments did not contribute to their investigation of functionally relevant proton movements.
2. The authors interpret the slower build-up of D85 sidechain C-H signals than D96 sidechain C-H signals as evidence for "delocalization" of the D85 sidechain proton, based on data in Supplementary Fig. 4. The authors should include plots of C-H crosspeak volumes or heights as a function of the CP contact time, with error bars in the plots, and they should quantify the difference in build-up rates. Their current interpretation is overly qualitative.
3. The meaning and significance of the phrase "the exchange peak of D85 Hd2 at the water frequency has a much higher intensity as the corresponding diagonal peak" is unclear. If the crosspeak is stronger than the diagonal peak, does this simply mean that the D85 sidechain proton is exchanging with multiple water protons during the exchange period of the pulse sequence?
4. Do the data for D85 simply mean that D85 is partially ionized under the conditions of these experiments?
5. The authors suggest that the cross peak between R82 and D85 may reflect exchange via water molecules 401 and 406. This seems speculative. Is there any information in the solid state NMR data that can be used to identify specific water molecules?
6. It seems to me that the rates of proton exchange are important. If these rates are small, then the observed exchange is less likely to be functionally relevant. Can the authors estimate the exchange rates from their data?
7. The authors seem to suggest that illumination is necessary for back exchange to occur. Has this been verified experimentally?
8. I do not understand the phrase "perpetuation of the so far observed exchange in the active site proton cage directly at the RSB".

Reviewer #2 (Remarks to the Author):

Overview:

Friedrich and co-workers report on water-protein proton chemical exchange, and the hydrogen-bonding network within the proton channel of bacteriorhodopsin (BR) in the dark-state. This intriguing characterization was performed using ^1H detected MAS solid-state NMR experiments and ab-initio molecular dynamics.

The major claims were:

- (1) "Reversible proton translocation" happens in the dark state and involves the RSB, a specific bound water molecule, D85, and R82
- (2) The dark state of the proton shuttle already has a "proton path" between D85 and R82
- (3) Retinal isomerization induced by light makes the exchange irreversible, leading to ^1H extracellular release.

Overall I think this is excellent work and wish the authors speedy revisions

Specific concerns

Line 31: I think proton "translocation" has a contextual meaning of "productive" transport between two compartments and is not appropriate here. I have interpreted the system to be in equilibrium; therefore, I think this is a simple chemical exchange. A chemical potential may have highly complicated the NMR experiments. I ask the equilibrated nature of the system to be clarified. Other groups have gone to great lengths to study these systems with a chemical potential or in the light state.

Line 39 and Line 250: While I agree that the retinal isomerization explanation for inducing irreversibility is logical, I am not sure it should be a conclusion. I don't believe you have direct evidence of this; you are in equilibrium. It seems to be a prediction that your model makes. Can you more clearly describe this as a prediction?

Can the authors distinguish the difference between chemical and spin exchange in the manuscript? Explicitly, under these proton concentrations, MAS rates, and distance are some of the cross-peaks explained by spin-exchange?

Other concerns:

Any details about the NMR data processing conditions are missing. Could you include processing parameters? I was curious because of the somewhat different peak shapes in Figure 2. In Figure 2D, the BR WT spectrum looks like it may have been processed with different apodization functions than the other spectra. Is this just due to signal-to-noise?

In Figure 2D, if the signal-to-noise difference is taken into account, is it still clear that the D85-H peak is absent in the mutant spectra?

Figure 2D: Does BR R82Q break the proton channel? The spectra indicate the water path is no-longer observable. Is this correct?

Line 130-131: This sentence is confusing. I don't understand what D85-H occupies multiple locations means. Are these multiple locations, i.e., different states if so I see no evidence of this or does this mean different peaks?

Your model does not evoke the exchange of the bound water molecule with the bulk water. Can you more clearly explain why there is a correlation at the water chemical shift for many of the residues?

Although the backbone assignments work is excellent, I don't think it was used to conclude anything. Can you clarify this? I don't suggest removal of the assignments, but can the chemical shift information be better used to show how the protein is folded? Perhaps, the authors can compare the chemical shifts to other NMR studies of BR? Additionally, there were no spectra shown for the HCO(CA)NH, (H)CBCANC, or (H)CBCA(CO)NH data sets mentioned in the methods section.

Mutations were used to make assignments that were important for proton dynamics. Can you add a discussion about how these mutations affect protein structure or function?

Does the ab Initio molecular dynamics show any indication of a bifurcated hydrogen bond (i.e., a 3 centered bond) between the carboxylic acid group and the water?

Can you include the hydrogen bond barrier energy in kcal/mol or kJ/mol for easier comparison?

Line 284: the authors state that the system was "equilibrated" under illumination for 4 hours at 15 C, is this long enough to reach equilibrium? Is there another literature precedent?

Line Supplementary Fig. 4: I think the CP build-up spectra shows that the dipolar coupling is stronger for D96-H/C. To claim the effect is due to proximity, at a minimum, I feel that you first need to show that the dynamics of the two groups are similar. The groups might have different $R_{1\rho}$ values, for example. Even then, dipolar coupling networks in partially deuterated protein are complicated. It might be helpful to show the time vs. intensity plot. I also can't determine the spinning rate or temperature you collected these data.

Other notes:

Line 474: "two glutamic acids(E194 and E194)", those are the same.

Figure 2e. The negative charge on the carboxylic acid group is closer to the carbon. Can you indicate the resonance structure with dashed double bonds or move the negative charge closer to the oxygen?

Figure 3a, 97 K: Spectra has sharp 'digital' horizontal bands running through the bulk peak. Is this is an artifact of processing? Alternatively, some experimental glitch?

Reviewer #3 (Remarks to the Author):

The manuscript by Friedrich et al. reports on exchange processes occurring in the dark-adapted state of bacteriorhodopsin (BR). Using ^1H detected solid-state NMR insights at very high resolution could be generated. Overall the presented data are of excellent scientific quality, both in terms of NMR spectroscopy as well as sample preparation and allow to provide new information about this frequently studied biological system.

Nevertheless, a few aspects should be addressed in a revised version:

Major aspects:

1. Data on RSB: While the data and its interpretation is very convincing for Fig. 1+2, the interpretation of the spectra in Fig. 3a is, in my opinion and at the current stage, not supported by the data. The presented spectra show that the two peaks representing the two retinal conformation of the dark-adapted state are only visible at low temperatures. The disappearance of these signals is interpreted as chemical exchange with H₂O 402. However, wouldn't the easiest explanation of the broadening in the ¹⁵N-¹H CP spectrum not be that the retinal undergoes a temperature dependent 13-cis+15-syn to 13-trans+15-anti exchange process in the ms time regime? Or it may just become overall more flexible/dynamic as compared to the rest of the protein. Is there a way to exclude these possibilities? If not, all current interpretations of the RSB need to be rewritten/removed from the manuscript. As a consequence, the term 'collective exchange process' (including the title) would probably also need to be changed.

2. CP-buildup: In general, it would be helpful to show a plot of the buildup behavior in addition to the spectra. Nevertheless, it is already clear from the provided data that there is a difference. However, the CP buildup behavior is determined by two main factors, (i) distance and (ii) dynamics. Without knowing one of these factors the other one cannot really be interpreted. The authors should include a potential variation in side-chain mobility as another possibility to explain the experimental data.

3. I did not get the point that connects the position (distance of 5Å) and the necessity that water 502 is highly dynamic. It may be helpful to have a more detailed discussion of this aspect.

Minor aspects:

4. Why do the wavelength-labels of the dark-adapted state differ from the ones reported by the Griffin/Herzfeld groups (who obtained the same ¹⁵N and ¹H chemical shifts)?

5. The term 'proton delocalization' should per definition refer to a nuclear quantum effect rather than fast changing positions of the hydrogen atom. It may therefore be helpful to include a short definition of the term and its meaning for the presented research at the beginning of the manuscript.

6. The terms 'light exposed'; and 'illuminated' is used several times in the manuscript to describe the sample state. In principle this is correct, but it may lead to some confusion since only the dark-adapted state was investigated. I would recommend to explain at the beginning why the sample was illuminated/light exposed and that all subsequent data was recorded on the dark-adapted state. To avoid misunderstandings the later (unnecessary) statement of illumination could be omitted.

Please find below our point-by-point responses. The comments of the reviewers are copied in blue and our answers in black, indented text. The relevant edited parts of the revised manuscript are provided in italic font (changes are highlighted in yellow) and specified by the new line numbers.

Reviewers' comments:

Reviewer #1 (Remarks to the Author):

This paper describes solid state NMR measurements on bacteriorhodopsin, designed to provide information about movements of protons within the bR proton-pumping channel that are relevant to bR function. The authors prepare triply-labeled (2H,15N,13C) purple membrane samples and allow limited back exchange of 2H to 1H, thus limiting the number of 1H solid state NMR signals and leading to relatively simple spectra in which certain crucial 1H signals can be resolved and assigned.

In principle, measurements of this type could provide important new information. However, I am not sure that the authors have succeeded in obtaining definitive new information, at least in part because the authors' description of their results is unclear. My overall evaluation is that this work is interesting and potentially important, but the authors need to state their conclusions more clearly and explain the experimental bases for these conclusions more succinctly.

In addition to addressing the specific points raised by reviewer #1, which helped us considerably to present the results more clearly (please see point-by-point responses below), we changed the following sentence in the abstract (lines 32–35) to emphasize that our results provide new information:

*'Here we demonstrate that reversible proton **displacement** happens already in the **equilibrated** dark-state of bacteriorhodopsin (BR), involving the retinal Schiff base (RSB), water 402, D85, and R82, **providing new information on proton exchange in BR.**'*

The results providing this new information are described specifically in the three sentences following this sentence in the abstract (lines 35–41).

Also, we have replaced 'translocation' by the term 'displacement' in this new sentence (as we did throughout the manuscript), as reviewers #2 (in his/her first point) and #3 (in his/her point 5) criticized using the terms 'translocation' and 'proton delocalization'. We agree with this, and therefore decided to use 'displacement' instead of 'translocation' or 'delocalization' to describe the observed changing positions of hydrogen atoms more accurately (see also the point-by-point answers to these specific concerns). In addition, 'equilibrated' is included in the sentence to address the first point of reviewer #2 (please see the respective point-by-point response below for further explanations).

Furthermore, we have moved the references from the abstract (lines 30 and 31) to the main text, in accordance with the submission guidelines for articles.

The authors should address the following points:

1. The description of the solid state NMR experiments begins with assignments of backbone 1H, 15N, and 13C signals from triple-resonance experiments. It seems that none of residues with assigned backbone signals participate in proton pumping. Is this true? If so, the authors should clarify that these triple-resonance experiments did not contribute to their investigation of functionally relevant proton movements.

Yes, it is true that none of the resonances that were assigned by triple-resonance experiments allowed us to assign functionally-relevant protons. This assignment was done to make sure that we know the origin of the respective signals, in order to not mistake them for signals in the Schiff base region or around the proton cage. To clarify this, we therefore inserted the following sentence in lines 114–117:

‘However, none of these assigned backbone resonances correspond to residues involved in proton pumping; we therefore applied the following strategy to study sites that contribute to proton displacements during BR molecular action.’

2. The authors interpret the slower build-up of D85 sidechain C-H signals than D96 sidechain C-H signals as evidence for "delocalization" of the D85 sidechain proton, based on data in Supplementary Fig. 4. The authors should include plots of C-H crosspeak volumes or heights as a function of the CP contact time, with error bars in the plots, and they should quantify the difference in build-up rates. Their current interpretation is overly qualitative.

This point was raised by all three reviewers. We have included the requested quantitative analysis (including standard deviations) and provide an edited Supplementary Figure (previously Supplementary Fig 4., now Supplementary Fig. 5) of the CP build-up for both D85 and D96. The Figure description reads now as follows:

‘Supplementary Fig. 5. Cross polarization (CP) build-up experiments of the D85 and D96 carboxyl group proton cross peaks. *Eight dipolar coupling-based, two-dimensional, proton-detected (H)COH spectra with varying CP contact times were recorded at 60 kHz MAS and an actual sample temperature of 291 K. The applied transfer time for both CP steps (^1H - ^{13}CO and ^{13}CO - ^1H) is indicated in each panel that show the eight experiments. The 200 μs -experiment (top left) is plotted with contours at the noise level, while all other spectra are plotted at the same contour levels (positive contours are shown in black and red for the D96- $\text{H}^{\delta 2}/\text{C}^{\gamma}$ and D85- $\text{H}^{\delta 2}/\text{C}^{\gamma}$ cross peaks, respectively, and negative contours are shown in grey).*

The build-up of the two cross peak intensities was analyzed quantitatively (bottom panel, the D96- $\text{H}^{\delta 2}/\text{C}^{\gamma}$ and D85- $\text{H}^{\delta 2}/\text{C}^{\gamma}$ data points are shown in black circles and red squares, respectively). We fitted the two intensities over the CP contact time with equation (1) using IGOR Pro Version 8.03 and the nonlinear least square method (shown in dashed lines):

$$I(t) = I_{\max}(1 - e^{-kt}); \quad k = \frac{1}{T_{\text{HC}}} \quad (1)$$

with the CP contact time t , cross peak intensity I , and the CP build-up time T_{HC} of the cross peak (k is the build-up rate). The obtained coefficient values \pm one standard deviation are given in the plot for both T_{HC} and I_{\max} for the D96- $\text{H}^{\delta 2}/\text{C}^{\gamma}$ and D85- $\text{H}^{\delta 2}/\text{C}^{\gamma}$ cross peaks. The D96- $\text{H}^{\delta 2}/\text{C}^{\gamma}$ cross peak ($T_{\text{HC}} = 310.79 \mu\text{s}$) builds up faster by a factor of about 2.3 than the D85- $\text{H}^{\delta 2}/\text{C}^{\gamma}$ signal ($T_{\text{HC}} = 692.71 \mu\text{s}$). Assuming that deuteration effects and ^{13}C - ^1H dipolar couplings, i.e. dynamics of the D96 and D85 carboxyl groups, are similar, this indicates that, comparing D96 and D85, the carboxylic acid proton is closer to the C^{γ} in the case of D96. However, potentially different side chain mobilities of the D85 and D96 residues may additionally contribute to the observed differences in CP build-up.’

In the main text, we have further edited the interpretation of the CP build-up experiments to take possible variations in side chain mobilities of D85 and D96 as an additional explanation for the observed differences into consideration (lines 157–160):

‘The D96 signal builds up faster than the D85 peak, indicating that the carboxylic acid proton in the D85 case is, on average, more distant from the C^{γ} and therefore closer to H_2O in comparison to D96. However, varying D85 and D96 side chain mobilities represent an additional source for the observed differences in CP build-up.’

3. The meaning and significance of the phrase "the exchange peak of D85 Hd2 at the water frequency has a much higher intensity as the corresponding diagonal peak" is unclear. If the crosspeak is stronger than the diagonal peak, does this simply mean that the D85 sidechain proton is exchanging with multiple water protons during the exchange period of the pulse sequence?

To address the question of involvement of multiple water protons and to explain our interpretation of the observed differences in intensities of the exchange cross peaks at the water frequency for the D96–H^{δ2} and D85–H^{δ2} protons, we have edited the respective statement as follows (lines 163–169):

‘Intensities of diagonal peaks in exchange spectra are directly correlated to the populations, whereas those of cross peaks are additionally modulated by the kinetics of the exchange process between the two nuclei giving rise to the cross peak as well as by the populations of their respective states. In line with the above-mentioned CP build-up experiments (Supplementary Fig. 5), the observed intensity differences therefore suggest that the localization of D85–H^{δ2} is in favor of the chemical environment of water, which could explain that it was not observed by vibrational spectroscopy so far.’

4. Do the data for D85 simply mean that D85 is partially ionized under the conditions of these experiments?

Yes, this is exactly how we interpret the spectra. We tried to explain this in the submitted manuscript already, for example in lines 119–121 of the initial manuscript, and aimed to emphasize it with Fig. 2e. To make it more clear and specific, we included the following sentence in the interpretation of the D85 data (lines 154–155):

‘D85 should therefore be considered as partially ionized under the applied experimental conditions.’

In the abstract, we have additionally edited the sentence about the findings concerning D85 accordingly (lines 35–38):

‘MAS NMR shows exchangeable protons at D85 and the RSB, the respective chemical shifts are in agreement with a carboxylic acid proton at D85 or a hydronium ion 402 as an equilibrium of tautomers, suggesting partial ionization of D85.’

5. The authors suggest that the cross peak between R82 and D85 may reflect exchange via water molecules 401 and 406. This seems speculative. Is there any information in the solid state NMR data that can be used to identify specific water molecules?

This is absolutely a valid and important point. Unfortunately, we cannot resolve different water populations via the proton frequencies at the obtained ¹H linewidths. It may be possible via ¹⁷O spectroscopy, however, very high magnetic fields (presumably above 1 GHz ¹H Larmor frequency) and a specific probe would be required. As we do not have access to such equipment, we cannot identify specific water molecules from our data. However, it is fairly obvious (since there are no other possibilities) that these two water molecules are involved in the observed proton exchange between R82 and D85 (in BR dark-state, there is no other possible direct connection for a proton path between R82 and D85 based on the X-ray structure solved at 100 K by Luecke et al., 1999). We have therefore edited the respective paragraph in the Results accordingly (lines 172–179):

‘The BR wildtype exchange spectrum shows a cross peak between R82 and D85–H^{δ2} at 6.2 ppm and 12.1 ppm ¹H chemical shift, respectively, which may reflect a signal due to long-range proton exchange between the carboxyl group of D85 and the guanidinium group of R82, potentially via the structurally close water molecules 401 and 406 (Figs. 1b, 2a,d and Supplementary Fig. 4)⁵. In general, we cannot distinguish different water populations spectroscopically, including bulk water and protein-bound water molecules. Thus, proton exchange between amino acids and the sum of all water molecules is observed. This may be resolved if smaller ¹H linewidths could be achieved by even higher MAS frequencies, or through ¹⁷O spectroscopy at very high magnetic fields.’

We have further edited the sentence about this in the Discussion (lines 277–281):

‘The detected protonated form of R82 and the exchange signal correlating R82–H^η and D85–H^{δ2} suggest chemical exchange between both residues within this network, presumably via the water

molecules 401 and 406 as these, based on the dark-state crystal structure solved at 100 K, may provide a direct connection via a hydrogen bond network (Figs. 2a,b,d and 1b)⁵.

6. It seems to me that the rates of proton exchange are important. If these rates are small, then the observed exchange is less likely to be functionally relevant. Can the authors estimate the exchange rates from their data?

As we apply ¹H-¹H chemical exchange spectroscopy (based on ZZ exchange), we probe exchange processes in the ms time regime (exchange in the order of several hundred μs may be detected as well). This means that the observed rates are indeed small, i.e. in the order of 100-10000 Hz (we assume the reviewer has the same conception of “small” in the context of proton exchange rates). (For comparison, the proton mean residence time between a water molecule and a carboxyl group computed in the *ab initio* molecular dynamics simulations performed in our study is about 1 ps, corresponding to a rate of $1 \cdot 10^{12}$ Hz, which we consider as “high”.)

However, this is a very important point by this reviewer as it addresses the conceptional design of our study and enables us to improve its description. As we measure the BR dark-adapted, resting state in its equilibrium, the rates underlying the observed proton exchange processes are not the rates of the proton exchanges during the photocycle, i.e. of the proton transport steps during transitions between photocycle intermediates. With the proton exchanges detected in our study, it is possible to conclude on preformed proton communication pathways between key sites, e.g. R82, D85, RSB and H₂O. This is also stated in the abstract (lines 32–42) and in the introduction (e.g. lines 76–90 and 93–97). Between the investigated key sites, however, different proton exchange rates may be observed in the BR dark-adapted state and during the photocycle. We therefore can (and do) not conclude on the proton transfer steps during the photocycle (which cannot be measured in equilibrium), but only on the proton exchange occurring in the BR dark-adapted state. This shows for the first time that reversible proton exchange happens already in the BR dark-state and reveals a proton cage in the active site of BR, thus showing (i) partial ionization of D85 (and hence a protonated state of D85 in the BR dark-adapted state which has never been observed), (ii) that a proton is displaced within the hydrogen bond between the RSB and H₂O 402, (iii) that R82 participates in the proton exchange network, (iv) an equilibrium of tautomers at H₂O 402 and a hydronium ion 402, and (v) that D96 exchanges protons with H₂O already in the resting state.

To still make more clear that we measure in equilibrium with (potentially) different exchange rates, we edited the following sentence in the introduction (lines 94–97):

It informs on exchange contributions of the chemical sites involved in the global proton relocation process of BR, even when investigating the dark-adapted equilibrium that potentially features different proton exchange rates compared to the transfer steps during the photocycle.

7. The authors seem to suggest that illumination is necessary for back exchange to occur. Has this been verified experimentally?

There is wealth of evidence since many decades and no doubt that the structural changes during the BR photocycle enable the vectorial proton transport. From the literature, it is clear that the sites involved, i.e. the side chains of D96, D85, R82, the retinal Schiff base and the structurally close water molecules undergo changes in their protonation states during the photocycle. As we have produced the purple membranes in deuterated form, we made sure that all sites involved in proton pumping are reprotonated by illuminating the samples prior to the NMR measurements, in accordance with procedures described in the literature. All observed NMR signals are in line with previous studies and are assigned by the mutations R82Q, D96N and D85T. As we did not explicitly verify the requirement of illumination for the BR photocycle experimentally, we therefore included the following sentence and literature references in lines 90–93:

'It is well-established that the structural rearrangements during the photocycle are required for complete proton transport through the BR pore from the uptake to the release site, which is exploited here for reprotonation of sites involved in the pathway^{9,23–25}.'

8. I do not understand the phrase "perpetuation of the so far observed exchange in the active site proton cage directly at the RSB".

We have rephrased this sentence for clarification (lines 194–197):

*'In agreement with the occurrence of a dynamic process (Fig. 2e), we see exchange broadening of the RSB proton or nitrogen signal (Fig. 3a), indicating **that** the so far observed exchange in the active site proton cage **involves** the RSB **NH moiety as well** (Figs. 1b and 2e).'*

Reviewer #2 (Remarks to the Author):

Overview:

Friedrich and co-workers report on water-protein proton chemical exchange, and the hydrogen-bonding network within the proton channel of bacteriorhodopsin (BR) in the dark-state. This intriguing characterization was performed using ¹H detected MAS solid-state NMR experiments and ab-initio molecular dynamics.

The major claims were:

- (1) "Reversible proton translocation" happens in the dark state and involves the RSB, a specific bond water molecule, D85, and R82
- (2) The dark state of the proton shuttle already has a "proton path" between D85 and R82
- (3) Retinal isomerization induced by light makes the exchange irreversible, leading to ¹H extracellular release.

Overall I think this is excellent work and wish the authors speedy revisions

Specific concerns

Line 31: I think proton "translocation" has a contextual meaning of "productive" transport between two compartments and is not appropriate here. I have interpreted the system to be in equilibrium; therefore, I think this is a simple chemical exchange. A chemical potential may have highly complicated the NMR experiments. I ask the equilibrated nature of the system to be clarified. Other groups have gone to great lengths to study these systems with a chemical potential or in the light state.

As stated above in our response to the first remark of reviewer #1, we have replaced "translocation" and 'delocalization' by 'displacement' throughout the manuscript. We agree with the reviewers that the two previously used terms may cause misunderstandings as they imply transport in the context of a chemical potential. Proton pumps, including BR, have been studied extensively in their activated states by other groups.

As we have measured BR in the equilibrated dark-state (as understood correctly by this reviewer), we have clarified this in the sentence (lines 32–35):

*'Here we demonstrate that reversible proton **displacement** happens already in the **equilibrated** dark-state of bacteriorhodopsin (BR), involving the retinal Schiff base (RSB), water 402, D85, and R82, **providing new information on proton exchange in BR.***

Line 39 and Line 250: While I agree that the retinal isomerization explanation for inducing irreversibility is logical, I am not sure it should be a conclusion. I don't believe you have direct evidence of this; you are in equilibrium. It seems to be a prediction that your model makes. Can you more clearly describe this as a prediction?

As we agree with the reviewer in this point, we have edited these two sentences (lines 41–42 and 321–324):

'We propose that retinal isomerization makes the observed proton exchange processes irreversible and delivers a proton towards the extracellular release site.'

'As proton pumping by BR is a directed process, we suggest that retinal isomerization makes the observed chemical exchange within the proton cage irreversible through disrupting the proton oscillation via tilting of the RSB, resulting in the extracellular release of a proton.'

Can the authors distinguish the difference between chemical and spin exchange in the manuscript? Explicitly, under these proton concentrations, MAS rates, and distance are some of the cross-peaks explained by spin-exchange?

Unfortunately, we cannot distinguish between these two exchange mechanisms. To clarify this, we have added the following sentence in lines 143–147:

'In general, the presented exchange spectra include contributions from chemical and spin exchange, however, we assume the latter to be of minor importance due to the distances between the diluted protons investigated and the high MAS frequencies applied, which significantly reduces spin diffusion based on dipolar couplings.'

Other concerns:

Any details about the NMR data processing conditions are missing. Could you include processing parameters? I was curious because of the somewhat different peak shapes in Figure 2. In Figure 2D, the BR WT spectrum looks like it may have been processed with different apodization functions than the other spectra. Is this just due to signal-to-noise?

We have included the processing parameters in the Methods section (lines 413–418):

'All spectra have been processed with Bruker Topspin Versions 3.4 or 4.0 and analyzed with CcpNmr Analysis Version 2.4. Gaussian (typically with a maximum at 0.04 and 40 Hz line broadening, except for the variable temperature series spectra for which a maximum at 0.1 and 100 Hz line broadening was used) and sine squared (sine bell shift of 2) apodization functions have been applied in the direct and indirect dimensions, respectively. For all spectra, baseline correction with a polynomial of degree 5 was used.'

The spectra in Figure 2 have been processed with the same parameters. We assume that the slightly different peak shapes of the BR wildtype spectrum are indeed due to differences in signal-to-noise. We have noticed in all experiments that the BR wildtype sample yielded better signal-to-noise than the mutants. This may be due to a higher stability of the 2D crystalline membrane patches of the native purple membrane resulting in higher CP-efficiencies and a higher active concentration of BR in the MAS rotor. As this remains speculative and is, at least in our opinion, not of further relevance for the interpretation of the data, we have refrained from including a statement about this in the manuscript.

In Figure 2D, if the signal-to-noise difference is taken into account, is it still clear that the D85-H peak is absent in the mutant spectra?

Yes, the D85-H^{δ2} is still missing, even when considering the different signal-to-noise ratios. We have included an explicit statement in lines 136–138 and additionally provide the new Supplementary Figure 4, which shows all mutant ¹H-¹H exchange spectra plotted down to the noise level to make this clear not only for D85, but also for D96 and R82:

'This is also confirmed by the missing diagonal and cross peaks at the D85–H^{δ2} frequency in the ¹H–¹H exchange spectrum of D85T, even when taking different signal-to-noise ratios into consideration (Fig. 2d and Supplementary Fig. 4).'

The caption of the new Supplementary Figure 4 reads as follows:

'Supplementary Fig. 4. ¹H–¹H exchange spectra of BR wildtype (wt, black spectrum), R82Q (green spectrum), D85T (orange spectrum) and D96N (blue spectrum). Negative contours are depicted in grey. The shown spectra are the same as in Fig. 2d, but the three mutant spectra are plotted at lower contours to visualize the noise level. This illustrates that the D85–H^{δ2}, D96–H^{δ2} and R82–H^γ protons are missing in the mutants, even when taking different signal-to-noise ratios into consideration.'

Figure 2D: Does BR R82Q break the proton channel? The spectra indicate the water path is no-longer observable. Is this correct?

Yes, this is correct and in line with previous studies (please see the description of the R82Q mutant and the literature references below).

We have included a more detailed description of the functional effects of the three mutants in lines 118–126, see also the other comment by this reviewer (four comments below):

'Protons in the proximity to the carboxyl groups of D96, D85, at the guanidinium group of R82, and involving nearby H₂O molecules were probed by a combination of two-dimensional ¹⁵N–¹H (Fig. 2a,b), ¹³C–¹H (Fig. 2a,c), both CP-based, and ¹H–¹H exchange spectroscopy (Fig. 2a,d) of BR wildtype purple membranes. We further investigated in this way BR mutated at the proton donor of the RSB (D96), the acceptor of the RSB proton (D85) and the proton shuttle R82: D96N, in which the reprotonation of the RSB is dramatically slowed down³², D85T that features a red-shift of the BR-absorption and complete inactivation of proton pumping^{33,34}, and R82Q, in which BR-absorption is red-shifted, the RSB-pKa lowered, and proton shuttling from D85 to the PRG severely hampered^{35,36}.'

Line 130-131: This sentence is confusing. I don't understand what D85-H occupies multiple locations means. Are these multiple locations, i.e., different states if so I see no evidence of this or does this mean different peaks?

We agree with the reviewer that we do not specifically detect multiple locations of this proton ('multiple' in the sense of more than two). The ¹H–¹H exchange spectra, the ¹³C–¹H spectra and the CP build-up experiments (Figs. 2a,c,d and Supplementary Figs. 4 and 5) however clearly show that D85–H^{δ2} is exchanged between H₂O and the D85 carboxyl group as we detect it at both sites. That means the proton is displaced between these two sites and thus occupies multiple (i.e. two different) locations (H₂O and the D85 carboxyl group) as described in our previous phrasing of the sentence. However, we have simplified this sentence to conclude more precisely and to avoid such confusion (lines 169–171):

'Collectively, our data provide evidence that D85–H^{δ2} is exchanged within the hydrogen bond between D85 and one of the H₂O molecules close to it (Figs. 1b, 2a,c,d,e and Supplementary Figs. 4 and 5).'

Your model does not evoke the exchange of the bound water molecule with the bulk water. Can you more clearly explain why there is a correlation at the water chemical shift for many of the residues?

For interpretation of our data and preparation of the initial version of the manuscript, we had taken this into consideration already. However, we failed to include a specific statement about it in the manuscript. We have therefore included two clarifying sentences about this in lines 175–179:

'In general, we cannot distinguish different water populations spectroscopically, including bulk water and protein-bound water molecules. Thus, proton exchange between amino acids and the sum of all

water molecules is observed. This may be resolved if smaller ^1H linewidths could be achieved by even higher MAS frequencies, or through ^{17}O spectroscopy at very high magnetic fields.'

In all cases where we conclude on proton exchange between amino acid side chains and water molecules (R82, D85 and D96), our model is still valid as we interpret all observed proton exchange based on the water molecules resolved in the crystal structure. For each of the three amino acids, the interpretation that protons are exchanged between these residues and specific H_2O molecules can still be made as there are no other possibilities for the protons to exchange with water, even when the sum of all water molecules is observed at the same ^1H chemical shift.

Although the backbone assignments work is excellent, I don't think it was used to conclude anything. Can you clarify this? I don't suggest removal of the assignments, but can the chemical shift information be better used to show how the protein is folded? Perhaps, the authors can compare the chemical shifts to other NMR studies of BR? Additionally, there were no spectra shown for the $\text{HCO}(\text{CA})\text{NH}$, $(\text{H})\text{CBCANC}$, or $(\text{H})\text{CBCA}(\text{CO})\text{NH}$ data sets mentioned in the methods section.

We appreciate this idea of the reviewer and included a more detailed interpretation of the backbone data and comparison to other NMR studies of BR in lines 111–117:

'The well-dispersed, amide group signals with a chemical shift pattern characteristic for α -helices clearly suggest that BR adopts its native seven-transmembrane helix fold (Supplementary Fig. 1). The ^{15}N - ^1H correlation and the observed $^{13}\text{C}\alpha$ and $^{13}\text{C}\beta$ chemical shifts agree well with previous NMR studies of purple membranes^{30,31}. However, none of these assigned backbone resonances correspond to residues involved in proton pumping; we therefore applied the following strategy to study sites that contribute to proton displacements during BR molecular action.'

We have additionally edited Supplementary Fig. 2 and included all six triple-resonance experiments (in addition to the previous $(\text{H})\text{CANH}$ and $(\text{H})\text{CA}(\text{CONH})$ experiments, also the $(\text{H})\text{CONH}$, $(\text{H})\text{CO}(\text{CA})\text{NH}$, $(\text{H})\text{CBCANH}$ and $(\text{H})\text{CBCA}(\text{CO})\text{NH}$ spectra are shown now as requested in this comment). (Note: We assume the requested $(\text{H})\text{CBCANC}$ is a typo by this reviewer and should be the $(\text{H})\text{CBCANH}$ dataset, and we included the $(\text{H})\text{CONH}$ for completion, even though not specifically asked for in this comment by the reviewer.)

The Figure caption of Supplementary Fig. 2 reads now as follows:

'Supplementary Fig. 2. Assignment of amide backbone signals in bacteriorhodopsin using triple-resonance magic angle spinning NMR experiments. As an example, sequential connections in two-dimensional strips of triple-resonance, three-dimensional $(\text{H})\text{CANH}$ (light blue), $(\text{H})\text{CA}(\text{CO})\text{NH}$ (black), $(\text{H})\text{CONH}$ (dark blue), $(\text{H})\text{CO}(\text{CA})\text{NH}$ (red), $(\text{H})\text{CBCANH}$ (positive contours (CA) in orange, negative contours (CB) in green) and $(\text{H})\text{CBCA}(\text{CO})\text{NH}$ (positive contours (CA) in magenta, negative contours (CB) in purple) spectra are shown for residues F71 to N76. Magnetization transfer pathways of the six experiments are schematically illustrated at the top. Dashed lines indicate correlations allowing the assignment.'

Mutations were used to make assignments that were important for proton dynamics. Can you add a discussion about how these mutations affect protein structure or function?

We have added a description of the functional and structural effects of the three mutants in lines 118–126:

'Protons in the proximity to the carboxyl groups of D96, D85, at the guanidinium group of R82, and involving nearby H_2O molecules were probed by a combination of two-dimensional ^{15}N - ^1H (Fig. 2a,b), ^{13}C - ^1H (Fig. 2a,c), both CP-based, and ^1H - ^1H exchange spectroscopy (Fig. 2a,d) of BR wildtype purple membranes. We further investigated in this way BR mutated at the proton donor of the RSB (D96), the acceptor of the RSB proton (D85) and the proton shuttle R82: D96N, in which the

reprotonation of the RSB is dramatically slowed down³², D85T that features a red-shift of the BR-absorption and complete inactivation of proton pumping^{33,34}, and R82Q, in which BR-absorption is red-shifted, the RSB-pKa lowered, and proton shuttling from D85 to the PRG severely hampered^{35,36}.

Does the *ab initio* molecular dynamics show any indication of a bifurcated hydrogen bond (i.e., a 3 centered bond) between the carboxylic acid group and the water?

We have included a new computation and analysis of the distances between the protons and oxygen atoms, which occur during the *ab initio* molecular dynamics simulation. The results are shown in the new Supplementary Fig. 6. This analysis reveals that the distances of each proton to the two oxygens of the same carboxyl group clearly differ, suggesting that bifurcation does not occur. This agrees well with the other results of the simulations. To include this finding and allow the reader to understand why such analysis of bifurcation is of relevance (given the nature and experimental design of our *ab initio* molecular simulations), we have therefore changed the description of the simulation results in lines 242–260:

*This model mimics the water molecule 402 that is coordinated by D212 and D85 (Fig. 1b) and thus enables an analysis of primary, thermally activated proton oscillations in the BR active site. The distances between the two carboxyl groups, and the positions of the central water molecule and of the protons are not fixed but are subject to thermal fluctuations. Depending on the distance between the carboxylic acid and water oxygens (RO_{cO_w}), the relative position of proton 2, described by an asymmetry coordinate ($RO_{cH}-RO_{wH}$, Fig. 3b), exhibits different behavior⁴⁵. Note that a negative asymmetry refers to the proton localized near the carboxyl group oxygen, and a positive asymmetry near the water oxygen. For RO_{cO_w} of 2.4–2.6 Å, intermediate positions of proton 2 are frequently observed, while for large values of RO_{cO_w} the proton tends to be localized near the central water molecule, as can be seen from the free energy distribution in Fig. 3c. This means that dynamic proton exchange is observed in time-resolved trajectories, from which a mean residence time of about 1 ps is determined (Fig. 3d). The analysis of the typical distances between the protons and oxygen atoms reveals that bifurcated, three-centered hydrogen bonds⁴⁶ do not appear in the model system (Supplementary Fig. 6). The proton transfer from the water oxygen to a carboxyl group involves a free-energetic barrier of about 3 $k_B T$ as a function of the asymmetry reaction coordinate $RO_{cH}-RO_{wH}$, as shown in Fig. 3e. In fact, Eckert and Zundel found a 6 $k_B T$ barrier for a fixed RO_{cO_w} of 2.65 Å from static *ab initio* SCF calculations⁴⁷. It is not surprising that the barrier height is smaller in our dynamic simulation which allows for kinetic relaxation of all positions.*

According to this new description of the simulations, we have edited the Figure caption of Fig. 3b,c,d,e (lines 626–635):

***b**, The proton distribution is analyzed by *ab initio* molecular dynamics simulations in a model system consisting of one H₂O molecule, two carboxyl groups and one excess proton which are all subject to thermal positional fluctuations. The distance between carboxyl group and water oxygen atoms (RO_{cO_w}) and the excess proton's relative asymmetry with respect to the two oxygens ($RO_{cH}-RO_{wH}$) are used as effective reaction coordinates. **c**, The free energy landscape of the proton is shown as a function of the coordinates, as defined in (b). **d**, Trajectories of the distances, RO_{cH} , of the three central hydrogens labelled as in (b) to the respective closest carboxyl group oxygen. The proton exchange is well visible as a fast jump process: Proton 1 resides near the closest carboxyl group oxygen for about 1 ps in the time interval from $t = 2.0$ ps to $t = 3.1$ ps. **e**, The free energy of the protons projected onto the asymmetry coordinate ($RO_{cH}-RO_{wH}$) indicates a low proton transfer barrier of about 3 $k_B T$.*

The Figure caption of the new Supplementary Fig. 6 reads as follows:

Supplementary Fig. 6. Analysis of occurring proton-oxygen distances between H₂O and carboxyl groups in *ab initio* molecular dynamics simulations. The model system used in this study consists of two carboxyl groups, one H₂O molecule and one excess proton (see also Fig. 3b). For each of the three protons (red, green and cyan), the shortest proton-oxygen distance (plotted in bright colors) is

computed for each of the nine NVE trajectories and compared to the distance of the same proton to the other oxygen (plotted in muted colors) in the respective carboxyl group. The distances among all protons vary strongly, providing evidence that bifurcated, i.e. three-centered, hydrogen bonds are not observed in the model system during the molecular dynamics simulations.'

Can you include the hydrogen bond barrier energy in kcal/mol or kJ/mol for easier comparison?

We have edited Fig. 3e accordingly and included the energy scale for a hydrogen bond barrier in kJ/mol at the right of the projected free energy plot.

Line 284: the authors state that the system was "equilibrated" under illumination for 4 hours at 15 C, is this long enough to reach equilibrium? Is there another literature precedent?

The reviewer seems to assume that during the illumination an equilibrium is established that is then measured by NMR. As this is not the case, we have described the experimental procedure during sample preparation more precisely in lines 362–375 and included the appropriate literature reference:

'The purple membranes were diluted in 90% / 10% $^1\text{H}_2\text{O}$ / $^2\text{H}_2\text{O}$ 50 mM Tris-HCl at pH 7.4 to 0.01 OD and illuminated (a 595 nm filter was used) for 4 hours under stirring in a water-cooled cuvette at 15 °C with a home-build illumination set-up. We thus assume that each bacteriorhodopsin molecule went through the photocycle during this photo-equilibration, thereby pumping protons and back-exchanging protons at key sites of the proton transport pathway (including the D85, D96 and R82 side chains, the retinal Schiff base and water molecules). The purple membranes were then collected by ultracentrifugation for 2 hours at 150,000 × g and 4 °C, and packed into 1.9 mm (for 20 kHz and 40 kHz MAS experiments) or 1.3 mm (for 60 kHz MAS experiments) Bruker MAS NMR rotors using home-made filling tools. Rotors were sealed with silicone rubber disks to avoid loss of liquid during MAS. Before starting acquisition of MAS NMR experiments, the samples were equilibrated in the dark (in the MAS NMR probe inside the magnet) for at least 1.5 hours to allow for establishing the BR dark-adapted state, i.e. the mixture of the 13-cis,15-syn and 13-trans,15-anti retinal configurations, which has been measured to reach equilibrium after 21 min⁴⁰.'

Line Supplementary Fig. 4: I think the CP build-up spectra shows that the dipolar coupling is stronger for D96-H/C. To claim the effect is due to proximity, at a minimum, I feel that you first need to show that the dynamics of the two groups are similar. The groups might have different R1rho values, for example. Even then, dipolar coupling networks in partially deuterated protein are complicated. It might be helpful to show the time vs. intensity plot. I also can't determine the spinning rate or temperature you collected these data.

Again, this point was raised by all three reviewers. As stated above in the point-by-point answer to the comment of reviewer#1, we have included the requested quantitative analysis in the new Supplementary Fig. 5. The Figure description (including the spinning frequency and temperature) reads now as follows:

'Supplementary Fig. 5. Cross polarization (CP) build-up experiments of the D85 and D96 carboxyl group proton cross peaks. Eight dipolar coupling-based, two-dimensional, proton-detected (H)COH spectra with varying CP contact times were recorded at 60 kHz MAS and an actual sample temperature of 291 K. The applied transfer time for both CP steps (^1H - ^{13}CO and ^{13}CO - ^1H) is indicated in each panel that show the eight experiments. The 200 μs -experiment (top left) is plotted with contours at the noise level, while all other spectra are plotted at the same contour levels (positive contours are shown in black and red for the D96- $\text{H}^{\delta 2}/\text{C}^{\gamma}$ and D85- $\text{H}^{\delta 2}/\text{C}^{\gamma}$ cross peaks, respectively, and negative contours are shown in grey).

The build-up of the two cross peak intensities was analyzed quantitatively (bottom panel, the D96- $\text{H}^{\delta 2}/\text{C}^{\gamma}$ and D85- $\text{H}^{\delta 2}/\text{C}^{\gamma}$ data points are shown in black circles and red squares, respectively). We fitted the two intensities over the CP contact time with equation (1) using IGOR Pro Version 8.03 and the nonlinear least square method (shown in dashed lines):

$$I(t) = I_{max}(1 - e^{-kt}); \quad k = \frac{1}{T_{HC}} \quad (1)$$

with the CP contact time t , cross peak intensity I , and the CP build-up time T_{HC} of the cross peak (k is the build-up rate). The obtained coefficient values \pm one standard deviation are given in the plot for both T_{HC} and I_{max} for the D96- $H^{\delta 2}/C^{\gamma}$ and D85- $H^{\delta 2}/C^{\gamma}$ cross peaks. The D96- $H^{\delta 2}/C^{\gamma}$ cross peak ($T_{HC} = 310.79 \mu s$) builds up faster by a factor of about 2.3 than the D85- $H^{\delta 2}/C^{\gamma}$ signal ($T_{HC} = 692.71 \mu s$). Assuming that deuteration effects and ^{13}C - 1H dipolar couplings, i.e. dynamics of the D96 and D85 carboxyl groups, are similar, this indicates that, comparing D96 and D85, the carboxylic acid proton is closer to the C^{γ} in the case of D96. However, potentially different side chain mobilities of the D85 and D96 residues may additionally contribute to the observed differences in CP build-up.'

As described in the last two sentences of the Figure caption of Supplementary Fig. 5, we also include side chain mobilities (i.e. different dipolar couplings) and deuteration effects as possible explanations, and we have further edited the interpretation of the CP build-up experiments in the main text (lines 157–160) as stated above in the answer to comment #2 by reviewer #1:

'The D96 signal builds up faster than the D85 peak, indicating that the carboxylic acid proton in the D85 case is, on average, more distant from the C^{γ} and therefore closer to H_2O in comparison to D96. However, varying D85 and D96 side chain mobilities represent an additional source for the observed differences in CP build-up.'

Other notes:

Line 474: "two glutamic acids(E194 and E194)", those are the same.

This has been corrected in line 602 to:

'Two glutamic acids (E194 and E204)'

Figure 2e. The negative charge on the carboxylic acid group is closer to the carbon. Can you indicate the resonance structure with dashed double bonds or move the negative charge closer to the oxygen?

We have changed this in Fig. 2e to the resonance structure with dashed double bonds as suggested by the reviewer.

Figure 3a, 97 K: Spectra has sharp 'digital' horizontal bands running through the bulk peak. Is this an artifact of processing? Alternatively, some experimental glitch?

This was indeed an artifact of processing. In the first version, there was accidentally a different degree of polynomial applied for baseline correction of the 97 K experiment as compared to the other ^{15}N - 1H correlation spectra of the variable temperature series shown in Fig. 3a. We have corrected it by applying the same degree of polynomial of 5 (all eight spectra are now processed with exactly the same processing parameters) and replaced the 97 K spectrum shown in the very left panel of Fig. 3a.

Reviewer #3 (Remarks to the Author):

The manuscript by Friedrich et al. reports on exchange processes occurring in the dark-adapted state of bacteriorhodopsin (BR). Using 1H detected solid-state NMR insights at very high resolution could be generated. Overall the presented data are of excellent scientific quality, both in terms of NMR spectroscopy as well as sample preparation and allow to provide new information about this frequently studied biological system.

Nevertheless, a few aspects should be addressed in a revised version:

Major aspects:

1. Data on RSB: While the data and its interpretation is very convincing for Fig. 1+2, the interpretation of the spectra in Fig. 3a is, in my opinion and at the current stage, not supported by the data. The presented spectra show that the two peaks representing the two retinal conformation of the dark-adapted state are only visible at low temperatures. The disappearance of these signals is interpreted as chemical exchange with H₂O 402. However, wouldn't the easiest explanation of the broadening in the ¹⁵N-1H CP spectrum not be that the retinal undergoes a temperature dependent 13-cis+15-syn to 13-trans+15-anti exchange process in the ms time regime? Or it may just become overall more flexible/dynamic as compared to the rest of the protein. Is there a way to exclude these possibilities? If not, all current interpretations of the RSB need to be rewritten/removed from the manuscript. As a consequence, the term 'collective exchange process' (including the title) would probably also need to be changed.

This is an important point, which we have not addressed in the interpretation of the RSB variable temperature spectra (Fig. 3a) in the initially submitted manuscript. While we agree with the reviewer that his/her ideas may represent in principle possible explanations under certain conditions, we are, however, convinced that such alternative data interpretation does not pertain in the specific case of our BR study for three reasons: (1) it is well-known that an exchange between 13-cis,15-syn and 13-trans,15-anti retinal is in the order of several minutes in BR dark-state without light irradiation (and not in milliseconds as suggested by the reviewer), (2) the active site, including the RSB does not show a high structural heterogeneity both in crystal structures and in NMR data (including ours) of BR dark-state, and (3) previous molecular dynamics simulations by the Elstner group support this view. We have therefore included the following sentences in the respective paragraph in lines 213–222:

'Another possible explanation for the disappearance of the two RSB proton signals could be conformational exchange between the 13-cis,15-syn and 13-trans,15-anti retinal configurations in the μs/ms time regime. If this would be the case, however, it would not take 21 min to reach equilibrium of the 40% / 60% distribution of these isomers as measured by Oesterhelt et al.⁴⁰. The low B-factors of the active site⁵, including the RSB, in the crystal structure and the well-defined, sharp NMR signals observed at low temperature further indicate high structural homogeneity. This excludes other conformational dynamics in the protein that could potentially lead to the observed line broadening, in agreement with previous studies employing molecular dynamics simulations by the Elstner group⁴¹.'

We certainly should have included this argumentation in the first version of the manuscript and thank the reviewer for raising this point. However, the wealth of evidence that allows to exclude structural dynamics of heavy atoms in the dark-state is a strong basis for our interpretation of RSB proton displacement. Therefore, the only possible conclusion for the obtained data in accordance with the current literature is the described RSB proton movement, in particular as it agrees well with the proton displacements observed at the other sites (D85 and H₂O 402, see Fig. 2e).

2. CP-buildup: In general, it would be helpful to show a plot of the buildup behavior in addition to the spectra. Nevertheless, it is already clear from the provided data that there is a difference. However, the CP buildup behavior is determined by two main factors, (i) distance and (ii) dynamics. Without knowing one of these factors the other one cannot really be interpreted. The authors should include a potential variation in side-chain mobility as another possibility to explain the experimental data.

Again, this point was raised by all three reviewers. As stated above in the point-by-point answers to the comments of reviewers #1 and #2, we have included the requested quantitative analysis in the new Supplementary Fig. 5. The Figure description reads now as follows:

'Supplementary Fig. 5. Cross polarization (CP) build-up experiments of the D85 and D96 carboxyl group proton cross peaks. Eight dipolar coupling-based, two-dimensional, proton-detected (H)COH spectra with varying CP contact times were recorded at 60 kHz MAS and an actual sample temperature of 291 K. The applied transfer time for both CP steps (¹H-¹³CO and ¹³CO-¹H) is indicated

in each panel that show the eight experiments. The 200 μ s-experiment (top left) is plotted with contours at the noise level, while all other spectra are plotted at the same contour levels (positive contours are shown in black and red for the D96-H^{δ2}/C^γ and D85-H^{δ2}/C^γ cross peaks, respectively, and negative contours are shown in grey).

The build-up of the two cross peak intensities was analyzed quantitatively (bottom panel, the D96-H^{δ2}/C^γ and D85-H^{δ2}/C^γ data points are shown in black circles and red squares, respectively). We fitted the two intensities over the CP contact time with equation (1) using IGOR Pro Version 8.03 and the nonlinear least square method (shown in dashed lines):

$$I(t) = I_{\max}(1 - e^{-kt}); \quad k = \frac{1}{T_{\text{HC}}} \quad (1)$$

with the CP contact time t , cross peak intensity I , and the CP build-up time T_{HC} of the cross peak (k is the build-up rate). The obtained coefficient values \pm one standard deviation are given in the plot for both T_{HC} and I_{\max} for the D96-H^{δ2}/C^γ and D85-H^{δ2}/C^γ cross peaks. The D96-H^{δ2}/C^γ cross peak ($T_{\text{HC}} = 310.79 \mu\text{s}$) builds up faster by a factor of about 2.3 than the D85-H^{δ2}/C^γ signal ($T_{\text{HC}} = 692.71 \mu\text{s}$). Assuming that deuteration effects and ¹³C-¹H dipolar couplings, i.e. dynamics of the D96 and D85 carboxyl groups, are similar, this indicates that, comparing D96 and D85, the carboxylic acid proton is closer to the C^γ in the case of D96. However, potentially different side chain mobilities of the D85 and D96 residues may additionally contribute to the observed differences in CP build-up.'

As described in the last two sentences of the Figure caption of Supplementary Fig 5, we also include side chain mobilities/dynamics (i.e. different dipolar couplings) as possible explanations, and we have further edited the interpretation of the CP build-up experiments in the main text as stated above in the answer to the comments by reviewers #1 and #2 concerning the quantitative CP build-up analysis that was missing in the initial version of the manuscript (lines 157–160):

'The D96 signal builds up faster than the D85 peak, indicating that the carboxylic acid proton in the D85 case is, on average, more distant from the C^γ and therefore closer to H₂O in comparison to D96. However, varying D85 and D96 side chain mobilities represent an additional source for the observed differences in CP build-up.'

3. I did not get the point that connects the position (distance of 5Å) and the necessity that water 502 is highly dynamic. It may be helpful to have a more detailed discussion of this aspect.

In the BR dark-state, there has been only one water molecule observed near the D96 carboxyl group so far (H₂O 502). As this is 5 Å away from D96, which is too far for a direct proton displacement and we still detect proton exchange between water and D96, H₂O 502 is required to be closer to the carboxyl group, at least occasionally, i.e. it may be dynamic. As we do not have direct evidence for this, we have toned down the interpretation and edited the discussion accordingly, including a more precise explanation (lines 296–303):

'The observed proton exchange between D96 and water in BR₅₆₈, however, requires H₂O molecules close to the carboxyl group already in the BR dark-state. The next water molecule, H₂O 502, occurs at a distance of 5.0 Å to the D96-O^{δ2} along the proton transport pathway towards the RSB, as measured in the crystal structure solved at 100 K (Supplementary Fig. 7)⁵. We thus suggest that, in the BR dark-state, water molecule 502 may be rather dynamic at room-temperature in contrast to the situation at cryogenic temperatures. The possibility of proton displacement between D96 and water molecules at the cytoplasmic entry site of the BR pore can be largely excluded due to low water accessibility in the dark-state⁴⁸.'

Minor aspects:

4. Why do the wavelength-labels of the dark-adapted state differ from the ones reported by the Griffin/Herzfeld groups (who obtained the same ¹⁵N and ¹H chemical shifts)?

All wavelength-labels already matched the ones used by the Griffin and Herzfeld groups, except BR₅₄₈ (Griffin and Herzfeld use BR₅₅₅ in their more recent publications instead). The BR₅₄₈ nomenclature is used for the 13-*cis*,15-*syn* form by many labs for a long time, including Sheves, Schulten, Lugtenburg, Herzfeld, Griffin, Rothschild, Mathies, Lewis and others.

For us, there is no particular point in using one of the two. As we are comparing the chemical shifts with the recent Griffin and Herzfeld NMR studies and to avoid confusion, we therefore have changed the wavelength-label of BR₅₄₈ to BR₅₅₅ throughout the manuscript (in Figs. 1c and 3a, Supplementary Table 2 and in lines 200, 205 and 624).

5. The term 'proton delocalization' should per definition refer to a nuclear quantum effect rather than fast changing positions of the hydrogen atom. It may therefore be helpful to include a short definition of the term and its meaning for the presented research at the beginning of the manuscript.

As both reviewers #2 and #3 asked to clarify the terms 'translocation' and 'proton delocalization', we think using 'displacement' instead of 'translocation' or 'delocalization' provides a more accurate description of our observations as we detect changing positions (chemical exchange) of hydrogen atoms. Therefore, a definition of 'proton delocalization' is not necessary anymore (it is not used anymore in the revised manuscript).

6. The terms 'light exposed'; and 'illuminated' is used several times in the manuscript to describe the sample state. In principle this is correct, but it may lead to some confusion since only the dark-adapted state was investigated. I would recommend to explain at the beginning why the sample was illuminated/light exposed and that all subsequent data was recorded on the dark-adapted state. To avoid misunderstandings the later (unnecessary) statement of illumination could be omitted.

In the introduction (in lines 86–90), we have now clarified that we measured the BR dark-state to observe protons at key sites of the proton transport pathway:

*'Direct proton detection was employed in a temperature range from 100 K to 290 K to monitor protons in the BR dark-adapted state (mixture of 13-*cis*,15-*syn* and 13-*trans*,15-*anti* retinal configurations) at key sites of the proton transport pathway, after introducing a subset of exchanging protons into the interior of the perdeuterated samples upon illumination.'*

In addition, this is stated in the abstract as well (lines 32–35):

'Here we demonstrate that reversible proton displacement happens already in the equilibrated dark-state of bacteriorhodopsin (BR), involving the retinal Schiff base (RSB), water 402, D85, and R82, providing new information on proton exchange in BR.'

In addition to the changes to address the reviewer's questions, we have edited the following:

- Title page: the affiliations (present addresses) have been updated
- Line 28: the sentence has been shortened to meet the maximum length of the abstract ('...and storage...' has been deleted)
- Lines 30 and 31: the references have been moved from the abstract to the main text
- Line 36: the sentence has been shortened to meet the maximum length of the abstract ('...in contact with D85 and at the RSB...' has been replaced by '...at D85 and the RSB...')
- Line 37: the sentence has been shortened to meet the maximum length of the abstract ('...hydrogen bonding to...' and '...part of...' have been deleted)
- Line 40: the sentence has been shortened to meet the maximum length of the abstract ('...the...', '...carboxyl groups show...' and '...molecules...' have been deleted)
- Line 44: the abbreviation for bacteriorhodopsin is introduced in the main text
- Line 49: the abbreviation for the retinal Schiff base is introduced in the main text
- Line 52: singular instead of plural was used by mistake (for non-consecutive steps)
- Line 54: the retinal configurations of BR₅₆₈ dark-state are described more accurately
- Line 57: the retinal configuration of L₅₅₀ is described more accurately
- Line 59: there was a typo (absorbtion vs. absorption)
- Line 77: it is specified that RSB, D85 and R82 form the active site proton cage
- Line 82: the abbreviation for magic angle spinning is introduced in the main text
- Line 86: we specify that native BR-enriched membranes are called purple membranes
- Lines 101–102: Heading and sub-heading is now used in the Results section
- Line 104–105: we describe that the NH spectra are based on through-space magnetization transfers
- Line 120: 'spectra' is replaced by 'spectroscopy'
- Lines 142–143: A sub-heading of this Results section is now provided and 'their' is included in the first sentence of this paragraph
- Line 143: a typo has been corrected (intriguing)
- Lines 172–173: the order of the chemical shifts is now correct
- Lines 183–184: the interpretation of R82 being an intermediate of the BR proton relocation pathway is now described more precisely
- Line 238: sub-heading is used for this Results paragraph
- Line 267: there was a typo (chemical shifts vs. chemical shift)
- Lines 272–274: Heading is now used for the Discussion, and we provide a new, introductory sentence to start the Discussion
- Lines 276–277: 'R82' and 'it' is used now in the correct order
- Line 315: a typo has been corrected (Intriguingly)
- Lines 320–321: the involvement of R82 in proton transport is described more accurately in accordance with our data
- Lines 450–452: the author contributions are updated to better reflect the performed work
- Line 458: the data availability statement is updated as the chemical shifts deposited in the BMRB will be released upon acceptance for publication of the article
- Line 465: a typo has been corrected (bacteriorhodopsin)
- Line 480: a typo has been corrected (M.L.)
- Line 490: a typo has been corrected (changes)
- Line 561: a typo has been corrected (bacteriorhodopsin)
- Line 591: a typo has been corrected (Cp2k)
- Line 594: a typo has been corrected (functional)
- Proton 'translocation' and 'delocalization/delocalized' are replaced by 'displacement/displaced' throughout the manuscript where appropriate (in line with the reviewer's comments)
- The references have been updated to meet the reviewers requests
- The numbering of Supplementary Figures has been changed

REVIEWERS' COMMENTS:

Reviewer #1 (Remarks to the Author):

The authors have made significant revisions that adequately address my comments on the original version of this manuscript. No further revisions are required.

Reviewer #2 (Remarks to the Author):

The authors satisfactorily addressed all issues that were raised.

Reviewer #3 (Remarks to the Author):

The authors provide a very thorough revision and adequately addressed all my points in the new version of the manuscript. I have no further comments and congratulate the authors for their nice work.

Manuel Etzkorn